# Towards Rule-Based Knowledge Sharing in Federated Learning

Zixuan Qin [1]  Qi Shen [2]  Liu Yang [2]  Qilong Wang [2]  Qinghua Hu [2]

## Abstract

Federated learning often faces both data and model heterogeneity, with the latter often more challenging. Architectural differences yield incompatible representation, making the knowledge-sharing carrier central to heterogeneous collaboration. Using proxy model enables distillation-based collaboration but incurs high communication and computation costs. Prototype-based carriers are lighter yet cause semantic confusion when incompatible features are mixed. Therefore, we propose rule-based federated learning (RFL) that shares interpretable, class-discriminative rules to enable heterogeneous collaboration, avoid feature confusion, and keep communication lightweight. RFL uses a rule network to unify clients' decision features and collaborates at the rule level, avoiding forcible averaging of incompatible representations. RFL selects sparse, high-coverage, beneficial rules for broadcasting, compressing shared knowledge into an interpretable class-rule set and reducing communication and computation costs. Each client selectively activates only rules relevant to its local classes, mitigating negative transfer while preserving personalization. Across heterogeneous settings, RFL achieves a better accuracy–communication trade-off.

## 1. Introduction

Federated learning (FL) is a collaborative learning paradigm with privacy preservation (McMahan et al., 2017). However, in real-world scenarios, such collaboration usually faces two types of heterogeneity: data (Lu et al., 2024; Huang et al., 2024) and model heterogeneity (Pei et al., 2024), with the latter often subsuming the former. Many personal-ized FL methods (Mildner et al., 2025; Zhou et al., 2026; Cheng et al., 2026) target data heterogeneity but typically assume homogeneous models, so they degrade in model-heterogeneous settings. More critically, heterogeneous models differ in representation semantics and structure, making what to communicate and how to learn across clients the key obstacle to practical FL deployment (Zhang et al., 2024a; Zhou et al., 2025).

Recent work in heterogeneous federated learning (HtFL) has largely sought to construct collaborative knowledge carriers across heterogeneous models (Li et al., 2025; Wu et al., 2024). (i) auxiliary proxy models trained on proxy data and distilled back to clients for indirect collaboration (Yi et al., 2024; Wu et al., 2022; Ma et al., 2025; Zhu et al., 2021). (ii) nested sub-models aggregated within a shared model family (Ilhan et al., 2023; Wu et al., 2024); and (iii) shared prototypes as the carrier, enabling feature aggregation and exchange (Tan et al., 2022; Zhang et al., 2024b; Zhou et al., 2025; Meng et al., 2026).

However, existing methods remain limited. Sub-model methods rely on within-family nesting and fail under substantially different client architectures. Auxiliary-model approaches centralize interaction into a black-box model and require proxy data and extra training, introducing privacy and deployment burdens while weakening interpretability. Prototype-based methods assume heterogeneous representations are comparable and additive; under architectural heterogeneity and mismatched semantic granularity, averaging compresses multi-scale cues into a single vector, causing feature mismatch and semantic confusion (Figure 1).

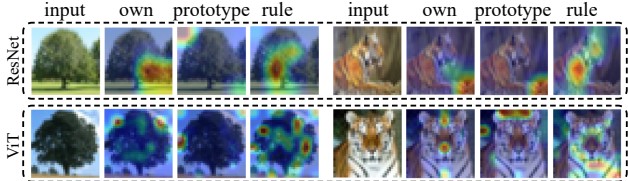

*Figure 1.* Semantic confusion of prototypes under strong heterogeneity. Own reflects each model's native feature response, while the averaged prototype yields inconsistent and incorrect attention across ResNet/ViT; in contrast, rule carrier preserves the correct focus across architectures.

Therefore, the key to HtFL is to *construct a shared knowledge carrier that is lightweight, interpretable, and exchange-*

[1]School of Computer Science and Technology, Tianjin University, Tianjin, China [2]School of Artificial Intelligence, Tianjin University, Tianjin, China. Correspondence to: Liu Yang <yangliuyl@tju.edu.cn>.

*Proceedings of the 43rd International Conference on Machine Learning*, Seoul, South Korea. PMLR 306, 2026. Copyright 2026 by the author(s).

*able across architectures, to mediate collaboration among clients.* Human individuals are inherently heterogeneous and mainly communicate through interpretable rule-based knowledge rather than raw perceptions (Garfield & Lew-Levy, 2025). Different individuals may acquire the same rule in different ways, allowing such rule knowledge to be transmitted stably (Antoun et al., 2025; Bähner et al., 2025). Inspired by this, we propose RFL, a rule-sharing-based mechanism for heterogeneous collaboration.

Instead of force-aligning heterogeneous features in a shared vector space, RFL treats client-uploaded class prototypes as feature evidence, and a server-side interpretable rule network absorbs these discriminative features to induce class-wise decision rules. Concretely, the rule network first binarizes feature vectors into a shared set of Boolean atoms. Then, within a unified rule space, it employs logical layers to induce class-wise decision rules that explicitly describe which AND/OR combinations of feature evidence constitute stable decision grounds for each class. Collaboration thus happens at the rule level, where different clients contribute complementary evidence, and the server consolidates it into a rule library that is shareable across architectures, avoiding the semantic mixing and feature distortion caused by averaging incompatible representations, as shown in Figure 1.

Building on this, we propose selective rule absorption. The server organizes rules by class, retains only high-coverage, positively contributing rules, and broadcasts them together with the rule network. Each client activates the corresponding rule constraints only for the classes it observes locally. This turns cross-client transfer from black-box matching into interpretable, selectable rule learning, mitigating negative transfer while preserving personalization. Furthermore, clients absorb rule knowledge jointly at the feature and logit levels, strengthening discriminative features that activate key rules and distilling the decision boundaries induced by the rule network. Our contributions are as follows:

- **Interpretable, lightweight knowledge carrier.** RFL uses interpretable class decision rules as the knowledge-sharing carrier, enabling heterogeneous clients to collaborate in a unified rule space while avoiding the semantic mismatch caused by averaging incompatible representations.

- **Controllable rule-based knowledge absorption.** RFL introduces selective rule absorption, broadcasting only high-coverage sparse rules. Clients absorb them at both the feature and logit levels, while keeping communication and computation lightweight.

- **Better accuracy–efficiency trade-off.** Experiments under heterogeneity show RFL achieves superior accuracy–communication–computation trade-off while keeping cross-architecture semantics.

## 2. Related Work

### 2.1. Heterogeneous Federated Learning

Recent personalized federated learning methods (M. Ghari & Shen, 2024; Ye et al., 2023; Qin et al., 2023; Mildner et al., 2025) mitigate data heterogeneity but are limited under architectural heterogeneity. Heterogeneous federated learning (HtFL) tackles model and data heterogeneity through cross-architecture knowledge carriers, including auxiliary proxy models, nested sub-models, and shared representations.

Auxiliary proxy-model methods (Du et al., 2025; Sun et al., 2025; Wang et al., 2024a) introduce an additional homogeneous model as an intermediary, enabling heterogeneous clients to transfer knowledge via distillation. Some methods distill a global student on the server using proxy data (Lin et al., 2020; Li et al., 2025), whereas FedGen (Zhu et al., 2021) and FedZGE (Ma et al., 2025) synthesize distillation data with a generator or zeroth-order logit feedback, respectively. FedKD (Wu et al., 2022) and FedMRL (Yi et al., 2024) train shared proxy models alongside local models on client data. FedSCE (Zhang et al., 2025b) further stabilizes transfer with subspace constraints. These methods often rely on proxy models, incur significant training overhead.

Nested sub-model approaches (Alam et al., 2022; Ilhan et al., 2023) enable parameter aggregation by training subnetworks within a shared backbone. Extensions further vary subnet widths or select submodels based on parameter importance (Diao et al., 2021; Wu et al., 2024). However, their reliance on a unified model family limits applicability to highly diverse, cross-family architectures.

Representation-based approaches (Zhang et al., 2024a; Qi et al., 2025; Wang et al., 2024b) avoid parameter averaging by exchanging intermediate class-level representations, most commonly class prototypes (Tan et al., 2022). FedTGP (Zhang et al., 2024b) further learns trainable global prototypes on the server via adaptive-margin contrastive learning, and FedSA (Zhou et al., 2025) introduces semantic anchors to encourage consistent representations. These methods treat exchanged representations as comparable across backbones, which can blur semantics and is hard to interpret under large architectural gaps. In contrast, we encode knowledge as class-wise rules, enabling interpretable collaboration.

### 2.2. Rule Learning and Reasoning

Rule learning and symbolic reasoning works most focuses on constructing interpretable logical structures within a single model. NLMs (Dong et al., 2019) and LNNs (Riegel et al., 2020) embed Horn clauses or weighted logic formulas via modular architectures and differentiable Boolean operators. Rule-based representation learners (Wang et al., 2021; Zhang et al., 2025a) parameterize classifiers as interpretable

discrete rules, optimized in continuous spaces. Building on these ideas, we treat client class prototypes as evidence samples and further induce class-level decision rules. Prior attempts to introduce rule reasoning into FL (Zhang & Yu, 2024; Xing et al., 2024) mainly target symbolic induction with limited scalability. In contrast, we introduce an independently trainable rule network as the collaborative knowledge carrier for cross-client rule learning.

# 3. Method

## 3.1. Problem Statement and Overall Objective

We consider a collaborative learning system with $N$ clients. Client $i$ holds a private dataset $D_i = \{(x, y)\}$, and all clients share the same label space $\mathcal{Y} = \{1, \dots, C\}$. The local model is written as $f_i(x) = h_i(g_i(x))$, where $g_i$ is the feature extractor and $h_i$ is the classifier. Let $\theta_i^g$ and $\theta_i^h$ denote their parameters, and $\theta_i = (\theta_i^g, \theta_i^h)$ be all trainable parameters of client $i$. The individual objective is

$$\mathcal{L}_{ce}^i(\theta_i) = \frac{1}{D_i} \sum_{(x,y) \in D_i} \ell_{ce}(f_i(x; \theta_i), y), \qquad (1)$$

where $\ell_{ce}$ denotes the cross-entropy loss. In homogeneous FL, a shared model $\theta$ is typically learned via parameter aggregation by solving $\min_\theta \sum_{i=1}^N \mathcal{L}_{ce}^i(\theta)$.

In the more realistic heterogeneous setting, client architectures differ, and direct parameter sharing is often infeasible. We abstract collaboration under heterogeneity into two coupled components: (i) a shareable knowledge carrier $\mathcal{K}$, and (ii) a mechanism $\Phi$ for absorbing collaborative knowledge from $\mathcal{K}$. Accordingly, the general objective of heterogeneous FL can be written as a joint optimization over self-learning and collaborative learning

$$\min_{\{\theta_i\}, \mathcal{K}, \Phi} \; J\big(\{F_i(\theta_i), \; \Phi(f_i(x; \theta_i), \mathcal{K})\}_{i=1}^N\big), \qquad (2)$$

where $J(\cdot)$ is a global generalization objective. Existing HtFL methods can be viewed as different instantiations of $(\mathcal{K}, \Phi)$, e.g., using class prototypes as $\mathcal{K}$ with feature-alignment constraints, or using a proxy model as $\mathcal{K}$ with knowledge distillation. To overcome their limitations, we instantiate $\mathcal{K}$ as a rule network $R$—an exchangeable, interpretable, and communication-efficient medium across architectures. We learn rule knowledge from client-uploaded feature prototypes and broadcast it back to support knowledge absorption during local training.

## 3.2. RFL Framework

### 3.2.1. LOCAL KNOWLEDGE EXTRACTION

To enable the rule network to absorb experiential knowledge from clients without accessing raw data, we require each client to upload class-level feature prototypes that summarize its local knowledge. During local training in round $t - 1$, the feature extractor $g_i$ of client $i$ outputs a representation vector of a unified dimension, denoted as $e_i^{t-1}(x) = g_i(x; \theta_i^g) \in \mathbb{R}^d$. For each class $c$, the class prototype of client $i$ is defined as

$$p_{i,c} = \frac{1}{|D_{i,c}|} \sum_{(x,y) \in D_{i,c}} e_i^{t-1}(x), \qquad (3)$$

where $D_{i,c}$ is the set of class-$c$ samples on client $i$. After local training, client $i$ uploads $(\{p_{i,c}\}_{c \in \mathcal{Y}}, A_i)$, where $A_i \in \mathcal{A}$ is its architecture label ($|\mathcal{A}| = A$), distinguishing class knowledge produced by different architectures.

### 3.2.2. RULE KNOWLEDGE CONSTRUCTION

The server aims to learn a global rule carrier $R$ as the basis for client collaboration. To this end, we adopt a rule-learning network (Wang et al., 2021) that compresses class prototypes collected across clients into logical rules for subsequent collaboration.

**Rule Training Data**. After receiving $\{(\{p_{i,c}\}_{c \in \mathcal{Y}}, A_i)\}_{i=1}^N$, the server converts these class prototypes into supervised samples for training the $R$. For each class $c$, the server collects the corresponding prototypes $\{p_{i,c}\}$ from all clients and groups them according to their architecture labels $A_i$. To prevent any single architecture from dominating rule learning, we perform balanced sub-sampling within each class $c$ across architectures, so that different architectures contribute comparably. To explicitly encode architectural differences within the same class, we concatenate $p_{i,c}$ with an architecture code to form a slot representation $s_{i,c} = [p_{i,c}; a_i] \in \mathbb{R}^{d+A}$, where $a_i \in \{0, 1\}^A$ is the one-hot encoding of the architecture $A_i$. Each pair $(s_{i,c}, c)$ is treated as a supervised sample, yielding a raw training set $D_{\text{raw}}$.

Relying only on class centers prototypes may cause the rule network to memorize center patterns rather than learn discriminative boundaries. To enhance boundary discrimination, we additionally generate a small number of boundary samples within each architecture by lightly interpolating between class prototypes and adding small perturbations. These samples are also concatenated with architecture codes and added to an augmented set $D_{\text{bnd}}$. The final server training set is $D_{\text{srv}} = D_{\text{raw}} \cup D_{\text{bnd}}$.

**Rule Network architecture**. The $R$ adopts an interpretable architecture. Prototypes are first discretized into Boolean atoms via the SoftBin module, then composed higher-order rules through logic layers, and finally mapped to classification logits, as illustrated in Figure 2. For clarity, we illustrate the rule learning process with a single training sample and write $s_{i,c} = [p_{i,c}; a_i]$ simply as $s = [p; a]$.

*SoftBin: architecture-conditioned discretization into*

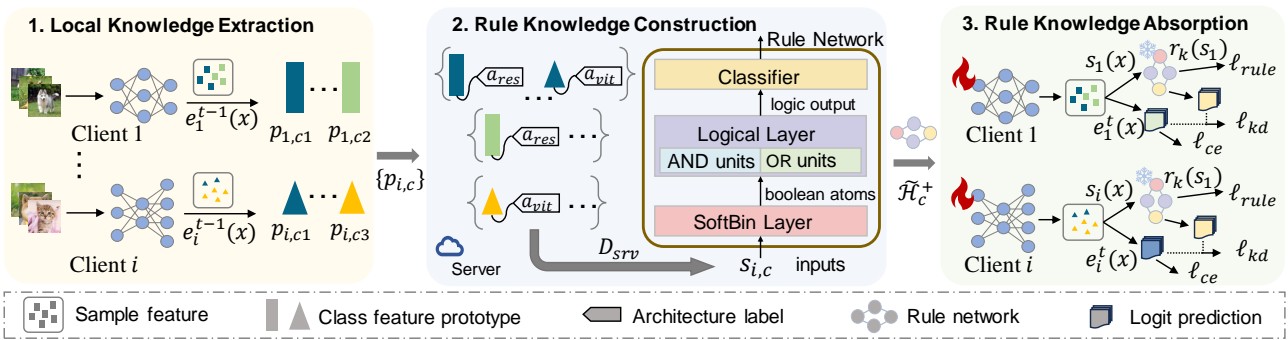

*Figure 2.* Overall framework of RFL. The entire framework consists of three main components: Stages 1 and 3 are performed on the client side, while Stage 2 is carried out on the server.

*Boolean atoms.* Given the input slot $s = [p; a]$, Soft-Bin first normalizes the prototype $p$ to reduce scale variations, producing $\hat{p} = \text{Norm}(p)$ computed per slot. It then applies an architecture-conditioned affine calibration $\tilde{p} = \gamma(a) \odot \hat{p} + \beta(a)$, where $\odot$ denotes element-wise multiplication, $\gamma(a)$ and $\beta(a)$ are learnable per-architecture calibration vectors selected by the one-hot code $a$. For each dimension $j$, we introduce $M$ learnable thresholds $\{th_{j,\tau}\}_{\tau=1}^{M}$ and generate Boolean atoms as

$$b_{j,\tau} = \mathbb{I}(\tilde{p}_j - th_{j,\tau} > 0), j = 1, \dots, d, \ \tau = 1, \dots, M, \quad (4)$$

where $\mathbb{I}(\cdot)$ is the indicator function. All atoms are flattened into $b = \text{vec}(\{b_{j,\tau}\}) \in \{0,1\}^{dM}$. Thus, prototypes from different architectures are mapped into a shared Boolean atom space, while architectural discrepancies are absorbed by $\gamma(a)$ and $\beta(a)$, avoiding direct averaging of semantically mismatched features. We concatenate $a$ back to the atom vector and feed $[b; a] \in \mathbb{R}^{dM+A}$ into subsequent logic layers to retain architecture-conditioned expressiveness.

*Logic layers: AND/OR composition of rule units.* The logic layers convert local threshold evidence into compositional class rules. While each SoftBin atom only indicates whether a calibrated coordinate crosses a learned threshold, AND/OR units model co-occurring and alternative evidence patterns, producing discriminative rules that clients can selectively absorb in a class-conditioned manner.

Let the input to the $l$-th logic layer be $u^{(l-1)} \in \{0,1\}^{d_{l-1}}$, where the SoftBin output serves as $u^{(0)}$. Each logic layer consists of two types of rule units: conjunctive (AND) units and disjunctive (OR) units. Their learnable connection weights are given by $W_{\wedge}^{(l)} \in [0,1]^{d_{l-1} \times n_{\wedge}}$ and $W_{\vee}^{(\ell)} \in [0,1]^{d_{\ell-1} \times n_{\vee}}$, where $n_{\wedge}$ and $n_{\vee}$ denote the numbers of AND and OR rule units, respectively. Each column of $W_{\wedge}^{(\ell)}$ (or $W_{\vee}^{(l)}$) corresponds to one rule unit, and each row corresponds to an input dimension from the previous layer.

To obtain discrete rule semantics, the forward pass applies

hardened connections by binarizing the weights as

$$\bar{W}_{\wedge}^{(l)} = \mathbb{I}(W_{\wedge}^{(l)} - 0.5 > 0), \qquad \bar{W}_{\vee}^{(l)} = \mathbb{I}(W_{\vee}^{(l)} - 0.5 > 0). \quad (5)$$

Accordingly, the $i$-th AND and OR rule units are defined as

$$o_{\wedge,i}^{(l)} = \bigwedge_{j: \bar{W}_{\wedge,ji}^{(l)}=1} u_j^{(l-1)}, \qquad o_{\vee,i}^{(l)} = \bigvee_{j: \bar{W}_{\vee,ji}^{(l)}=1} u_j^{(l-1)}, \quad (6)$$

where $\bar{W}_{\wedge,ji}^{(l)} = 1$ (or $\bar{W}_{\vee,ji}^{(l)} = 1$) indicates that the $j$-th input from the previous layer participates in the $i$-th AND (or OR) rule unit. The output of the $l$-th logic layer is obtained by concatenating all AND and OR rule unit outputs, $u^{(l)} = o_{\wedge}^{(l)} \oplus o_{\vee}^{(l)}$, where $\oplus$ denotes vector concatenation. Since the architecture code is fed into the logic layers and the rule structure is shared across architectures, the network can use $a$ to activate different rule combinations for inputs from different architectures, enabling knowledge fusion.

*Shared classifier.* Given the final logic output $r(s) = u^{(L)} \in \mathbb{R}^{d_L}$, the rule network employs a shared linear classifier to produce class logits for all architectures

$$z^R(s) = W_{\text{cls}} \, r(s), \quad (7)$$

where $W_{\text{cls}} \in \mathbb{R}^{C \times d_L}$ is the classification weight matrix. Therefore, $R$ learns cross-client decision rules in a shared rule space, enabling collaboration at the rule level.

**Rule Network Training.** The server trains the $R$ on the dataset $D_{\text{srv}}$. Let $\theta_R$ denote all trainable parameters of the rule network. The server-side objective minimizes the standard multi-class cross-entropy

$$\min_{\theta_R} \ \mathcal{L}_R = \frac{1}{|D_{\text{srv}}|} \sum_{(s,c) \in D_{\text{srv}}} \ell_{\text{ce}}\big(z^R(s; \theta_R), c\big), \quad (8)$$

Training relies only on lightweight prototype samples and does not require any additional real data. Since the inputs are low-dimensional prototypes rather than raw images, the $R$ can be implemented with only a few logic layers, together

with SoftBin and a linear classifier, resulting in a compact model with low computational cost.

After convergence, we extract class-specific positive rules from the shared classifier $W_{\text{cls}} = [w_1^\top; \ldots; w_C^\top]$, where $w_c \in \mathbb{R}^{d_L}$ corresponds to class $c$. We define the positively contributing rule indices as $\mathcal{H}_c^+ = \{ v \in \{1, \ldots, d_l\} \mid w_{c,v} > 0 \}$, and rank $\mathcal{H}_c^+$ by $w_{c,v}$ in descending order. The sign and magnitude of $w_{c,v}$ indicate how strongly the rule supports class $c$. However, not all positive rules are equally reliable. Since high-quality rules should be sparse yet high-coverage, we broadcast only a pruned subset $\tilde{\mathcal{H}}_c^+ \subset \mathcal{H}_c^+$ that contains the top-$\pi$ ranked positive rules. We start from a small head set in early rounds and gradually enlarge it over global rounds, exposing high-confidence class-relevant evidence first and enriching the rule set as training stabilizes.

Finally, the server broadcasts the $\theta_R$ and $\{\tilde{\mathcal{H}}_c^+\}_{c=1}^C$ to all clients. Compared to sharing full model parameters or raw features, this transmission involves only a small number of parameters and rule indexes, yielding an interpretable yet lightweight communication payload.

### 3.2.3. RULE KNOWLEDGE ABSORPTION

After receiving rule knowledge at round $t$, each client instantiates $\Phi$ by absorbing cross-client information via rule-based constraints on the relevant classes. For a single sample $(x, y)$ on client $i$, the local model first produces the logits $z_i(x) = h_i(g_i(x))$ and feature representation $e_i^t(x)$ via the feature extractor $g_i$ and $h_i$. Then concatenate $e_i^t(x)$ with the architecture one-hot code $a_i$ to form a slot representation $s_i(x) = \big[e_i^t(x); a_i\big] \in \mathbb{R}^{d+A}$, which is fed into the $R$:

$$\big(z^R(s_i(x)), r(s_i(x))\big) = R(s_i(x)). \qquad (9)$$

Here $z^R(s)$ denotes the rule-network logits and $r(s)$ denotes the activation vector of logic units. We freeze the received rule-network parameters $\theta_R$ during local training, using $R$ as a lightweight inference module with stable semantics. The client then optimizes $\theta_i$ under two constraints imposed by $R$.

**Rule constraint: class-conditioned rule activation.** For a sample with label $c$, we impose constraints only on the positive-rule set of class $c$, thereby encouraging the local representation to yield feature evidence that sufficiently activates these class-relevant logic units. Let $r_k(s_i)$ be the activation of the $k$-th logic unit for $(s_i, c)$. We define a margin-based soft hinge rule loss as

$$\ell_{\text{rule}}(s_i, y = c) = \frac{1}{|\tilde{\mathcal{H}}_c^+|} \sum_{k \in \tilde{\mathcal{H}}_c^+} \Big[ \max\big(0, \, m - r_k(s_i)\big) \Big]^2, \qquad (10)$$

where $m \in (0, 1)$ is a fixed margin. A penalty is incurred only when a positive rule is activated below $m$; well-satisfied rules receive no additional pressure. By

activating only a small set of class-relevant rules, each client avoids constraints from noisy or locally incompatible rules, enabling controllable knowledge absorption and mitigating negative transfer. Aggregated over the local dataset $D_i$, the corresponding rule loss is $\mathcal{L}_{\text{rule}}^i = \mathbb{E}_{(x,y) \sim D_i}\big[\ell_{\text{rule}}(s_i(x), y)\big]$.

**Rule distillation: aligning logits with rule predictions.** We further use $z^R(s)$ as an additional set of *teacher logits*, interpreting it as class scores distilled from cross-client prototypes. We introduce a standard temperature-based distillation loss on softened distributions:

$$\ell_{\text{kd}}(x, y) = \text{KL}\Big(\text{softmax}\Big(\tfrac{z^R(s(x))}{T_{tp}}\Big) \, \big\| \, \text{softmax}\Big(\tfrac{z_i(x)}{T_{tp}}\Big)\Big), \qquad (11)$$

where $T_{tp} > 0$ is the temperature. To mitigate scale discrepancies across architectures and task heads, we apply simple standardization to the teacher logits in practice.

Intuitively, this converts the global decision boundary learned by the rule network in the prototype space into a probability distribution that the local model can imitate, enabling a soft alignment from rules to local predictions. The corresponding $i$-th client distillation loss is $\mathcal{L}_{\text{kd}}^i = \mathbb{E}_{(x,y) \sim D_i}\big[\ell_{\text{kd}}(x, y)\big]$.

**Overall local objective.** Equivalently, the local optimization can be summarized as

$$\mathcal{L}_{\text{local}}^i = \mathcal{L}_{\text{ce}}^i + \alpha_{ru} \, \mathcal{L}_{\text{rule}}^i + \alpha_{kd} \, \mathcal{L}_{\text{kd}}^i, \qquad (12)$$

where $\alpha_{ru}, \alpha_{kd} \geq 0$ monotonically increasing with the global round $t$. After local updates, each client re-estimates its class prototypes $p_{i,c}$ and uploads the updated class-level summaries to the server for the next round of rule learning and collaboration. Overall, by instantiating $\Phi$ as $\mathcal{L}_{\text{rule}} + \mathcal{L}_{\text{kd}}$, clients absorb rule knowledge from the shared carrier $R$ in an interpretable and lightweight manner, enabling heterogeneous collaborative learning.

### 3.3. Theoretical Analysis

**Definition 3.1** (Architecture-invariant rule space)**.** Define the rule hypothesis class $H_R = \{\varphi(r) = W_{\text{cls}} r \mid W_{\text{cls}} \in \mathbb{R}^{C \times d_L}\}$. Let $\mathcal{S}$ denote the set of slots. All client models are embedded into the common rule space $\mathcal{R} = \Pi(\mathcal{S})$ and the rule network $R$ is of the form $R = \varphi \circ \Pi$ for some $\varphi \in H_R$. For any two slots $s, s' \in \mathcal{S}$, $\Pi(s) = \Pi(s') \implies \varphi(\Pi(s)) = \varphi(\Pi(s'))$ for all $\varphi \in H_R$. In particular, backbone-specific differences are absorbed by the architecture-conditioned SoftBin mapping in Eq. (4), while all collaborative decisions are taken in the shared rule hypothesis class $H_R$.

**Lemma 3.2** (Selective absorption in the rule space)**.** *For any margin buffer $\eta \in (0, m)$ and any slot $s$ with label $c$, define the set of severely under-activated positive rules $V_c(s, \eta) = \big\{k \in \tilde{\mathcal{H}}_c^+ \mid r_k(s) \leq m - \eta\big\}$. Then the rule*

*Table 1.* Classification accuracies (%) and communication costs (MB/round) under different data partitions. Supv. Gran. denotes *Collaborative Supervision Granularity*: RS (*Real-Sample*) trains and aggregates proxy models on clients' real local data; SS (*Synth-Sample*) learns from synthetic/query samples; CS (*Class-Summary*) transfers class-level summaries.

| Method Group | Supv. Gran. | Methods | Cifar10 | | | | Cifar100 | | | | Tiny-Imagenet | | | |
| --- | --- | --- | --- | --- | --- | --- | --- | --- | --- | --- | --- | --- | --- | --- |
| | | | | Acc.↑ | | Comm.↓ | | Acc.↑ | | Comm.↓ | | Acc.↑ | | Comm.↓ |
| | | | Pat | $Dir_{0.1}$ | $Dir_{0.3}$ | | Pat | $Dir_{0.1}$ | $Dir_{0.3}$ | | Pat | $Dir_{0.1}$ | $Dir_{0.3}$ | |
| Proxy-model methods | SS | FedGen | 83.99 | 82.12 | 74.21 | 46.99 | 58.76 | 54.35 | 37.85 | 62.84 | 40.64 | 32.85 | 26.34 | 80.44 |
| | RS | FedMRL | 84.75 | 83.35 | 75.50 | 260.11 | 63.24 | 58.43 | 44.06 | 265.36 | 42.58 | 33.49 | 23.79 | 271.26 |
| | SS | FedZGE | 82.73 | 79.41 | 71.02 | 993.52 | 58.68 | 55.10 | 38.32 | 1025.16 | 41.39 | 33.35 | 24.42 | 4030.31 |
| | RS | FedSCE | 85.22 | 84.62 | 77.07 | 260.11 | 63.62 | 57.25 | 44.53 | 265.36 | 43.52 | 36.13 | 26.57 | 271.26 |
| Prototype methods | CS | FedProto | 84.18 | 81.56 | 73.71 | 0.76 | 58.92 | 54.52 | 38.54 | 6.74 | 39.45 | 33.11 | 23.42 | 13.48 |
| | CS | FedTGP | 83.71 | 82.39 | 74.53 | 0.76 | 59.37 | 55.63 | 39.34 | 6.74 | 40.36 | 33.83 | 23.82 | 13.48 |
| | CS | FedSA | 83.89 | 82.69 | 73.98 | 0.76 | 57.81 | 53.14 | 38.30 | 6.74 | 40.74 | 32.74 | 23.12 | 13.48 |
| Ours | CS | RFL | 86.14 | 83.80 | 77.10 | 36.7 | 63.02 | 58.25 | 43.80 | 41.98 | 43.87 | 36.92 | 27.41 | 46.24 |

*loss $\ell_{\text{rule}}(s, y = c)$ in Eq. (10) controls both the fraction of severely violated positive rules and the total activation of the positive-rule set: $\frac{|V_c(s,\eta)|}{|\tilde{\mathcal{H}}_c^+|} \leq \frac{\ell_{\text{rule}}(s,y=c)}{\eta^2}$, $\sum_{k \in \tilde{\mathcal{H}}_c^+} r_k(s) \geq |\tilde{\mathcal{H}}_c^+| (m-\eta)\left(1 - \frac{\ell_{\text{rule}}(s,y=c)}{\eta^2}\right)$. In particular, if $\ell_{\text{rule}}(s, y = c) \leq \Lambda \ll \eta^2$, then only a small fraction of positive rules can stay far below $m - \eta$. Most class-positive rules are sufficiently activated in the shared rule space.*

**Assumption 3.3** (Positive-rule weight structure). As in the rule network definition, $r(s) \in [0,1]^{d_L}$ and $z_c^R(s) = \sum_{k=1}^{d_L} w_{c,k} r_k(s)$. For any sample $(s, y)$, we define the rule-space margin as $\text{margin}(s, y) = z_y^R(s) - \max_{c' \neq y} z_{c'}^R(s)$. We assume there exist constants $\alpha_{\text{pos}} > 0$ and $B_w > 0$ such that, for all classes $c$ and all indices $k \in \{1, \ldots, d_L\}$: (i) if $k \in \tilde{\mathcal{H}}_c^+$ then $w_{c,k} - \max_{c' \neq c} w_{c',k} \geq \alpha_{\text{pos}}$; (ii) $\|w_c\|_2 \leq B_w$.

**Lemma 3.4** (Rule-space margin lower bound). *For each class $c$, let $K_c^+ = |\tilde{\mathcal{H}}_c^+|$ and $K_{\min}^+ = \min_c K_c^+$. Under Assumption 3.3 and $r_k(s) \in [0,1]$, for any $\eta \in (0, m)$ and any sample $(s, y = c)$ satisfying $\ell_{\text{rule}}(s, y = c) \leq \Lambda$, the rule-space margin satisfies $\text{margin}(s,y) \geq \Gamma(\Lambda, \eta)$, where $\Gamma(\Lambda, \eta) = \alpha_{\text{pos}} K_{\min}^+ (m-\eta)\left(1 - \frac{\Lambda}{\eta^2}\right) - 2B_w\sqrt{d_L}$.*

**Theorem 3.5** (Rule-regularized local generalization). *Fix a client $i$ and $\eta \in (0, m)$, and let $\Gamma(\Lambda_i, \eta)$ denote the rule-space margin lower bound from Lemma 3.4 for samples with $\ell_{\text{rule}} \leq \Lambda_i$. Let $D_i = \{(x_j, y_j)\}_{j=1}^{n_i}$ be drawn i.i.d. from $\mathcal{D}_i$, where $n_i = |D_i|$. Denote by $L_i(f_i)$ the 0–1 risk of the local model $f_i$ on $\mathcal{D}_i$. Then there exists a universal constant $C_1 > 0$ such that, with probability at least $1 - \delta$ over the draw of $D_i$,*

$$L_i(f_i) \leq C_1 \frac{B_w\sqrt{d_L}}{\Gamma(\Lambda_i, \eta)\sqrt{n_i}} + \varepsilon_i^{\text{align}} + \sqrt{\frac{\log(1/\delta)}{2n_i}}, \quad (13)$$

*where $\varepsilon_i^{\text{align}} := \Pr_{x \sim \mathcal{D}_i}\left[\hat{y}_i(x) \neq \hat{y}^R(s_i(x))\right]$ is the alignment error between the local task head $f_i$ and the shared rule classifier $\varphi$ on client $i$.*

**Remark.** Theorem 3.5 makes explicit how the rule-regularized objective controls each client's generalization. The leading term $\frac{B_w\sqrt{d_L}}{\Gamma(\Lambda_i,\eta)\sqrt{n_i}}$ is entirely defined in the shared rule space, driving the local rule loss $\ell_{\text{rule}}$ below $\Lambda_i$ enlarges the margin lower bound $\Gamma(\Lambda_i, \eta)$ and therefore shrinks this complexity term, which depends only on the rule-network output dimension $d_L$ and the weight-norm bound $B_w$. The second term $\varepsilon_i^{\text{align}}$ measures how well the local task head $f_i$ matches the shared rule classifier. Distillation reduces this mismatch, so controlling $\varepsilon_i^{\text{align}}$ lets $L_i(f_i)$ benefit from the rule-space generalization of the architecture-invariant rule classifier. The last term is a standard concentration term depending on the sample size $n_i$. Overall, RFL improves local generalization by enforcing class-consistent rule activation in a shared rule space and distilling the resulting architecture-invariant decision semantics back into each heterogeneous client. Details are in Appendix C.

## 4. Experiments

### 4.1. Experimental Setup

**Datasets.** We evaluate RFL on CIFAR-10, CIFAR-100 (Krizhevsky et al., 2009) and Tiny-ImageNet (Deng et al., 2009) datasets. To characterize data heterogeneity, we adopt two commonly used partitioning schemes: a pathological setting and a practical setting. In the pathological setting, following FedAvg (McMahan et al., 2017), each client is randomly assigned non-overlapping and imbalanced samples from 3/15/30 classes on CIFAR-10/CIFAR-100/Tiny-ImageNet, respectively. In the practical setting, following (Li et al., 2021), we construct a Dirichlet non-IID split by sampling class proportions from $\text{Dir}(\alpha)$ for each client and allocating data accordingly, where $\alpha$ is typically set to 0.1.

**Baselines.** We compare RFL with popular HtFL methods, including FedProto (Tan et al., 2022), FedGen (Zhu et al., 2021), FedTGP (Zhang et al., 2024b), FedMRL (Yi et al., 2024), FedZGE (Ma et al., 2025), FedSA (Zhou et al., 2025)

and FedSCE (Zhang et al., 2025b).

**Model.** The heterogeneity of the models mainly lies in the choice of feature extractors. For tasks of different difficulty, we adopt different heterogeneous model configurations. On CIFAR-10 we use LeNet, AlexNet, and ResNet10; on CIFAR-100 we use AlexNet, ResNet10, and ViT-B/16; and on Tiny-ImageNet we use MobileNet, ResNet18, and ViT-B/16.

The common feature dimension is set by unifying the last representation layer of each backbone, without adding an RFL-specific projection head. This provides a shared prototype interface rather than reducing model heterogeneity. Prototype-based baselines use the same interface, while proxy-model baselines keep the same backbones and training budget.

**Implementation Details.** The experiments use 30 clients; each round runs 3 local epochs with Adam (lr=0.0003). The number of global rounds is set per dataset and rule network is optimized with Adam (lr=0.02). Each client splits its local data into 83.3%/16.7% train/test, and we report the average test accuracy over all clients. Additional details are provided in the Appendix.

### 4.2. Comparison to State-of-the-Arts

**Performance.** Table 1 shows that RFL consistently outperforms prototype-based baselines, with larger gains in more challenging settings; the maximum improvement exceeds 4.0%. When cross-architecture representations are semantically misaligned, prototype aggregation mixes incompatible semantics, whereas RFL transfers migratable discriminative knowledge via a shared rule space, leading to more effective collaboration. FedMRL/FedSCE achieve high accuracy by aggregating locally updated auxiliary models into a global teacher trained with sample-level supervision from all client data. This effectively approximates centralized training but requires much higher communication cost.

**Accuracy–Efficiency Trade-off.** As reported in Table 1, compared to proxy-model approaches, RFL reduces the average communication cost by 79.0% across all tasks while achieving comparable or higher accuracy, despite relying only on compact class-level rule as the knowledge-sharing carrier. Figure 3 further indicates that RFL lies in the "low-cost, high-accuracy" region and remains competitive with the strongest baselines. This advantage stems from using a lightweight rule network, avoiding repeated transmission and local training of black-box auxiliary models while preserving interpretable rule-based decision knowledge.

### 4.3. Rule Interpretability

**Rule sparsity and coverage.** Meaningful rule sets should be sparse yet high-coverage, with correct decisions relying

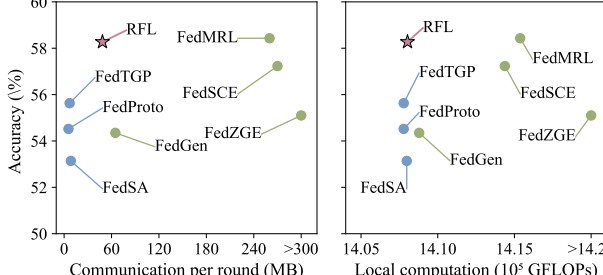

*Figure 3.* Accuracy–efficiency trade-offs (CIFAR-100). Blue: prototype-based; green: proxy-model-based. Communication cost is the total bidirectional traffic between server and clients per round. Computation is the total local training cost per round.

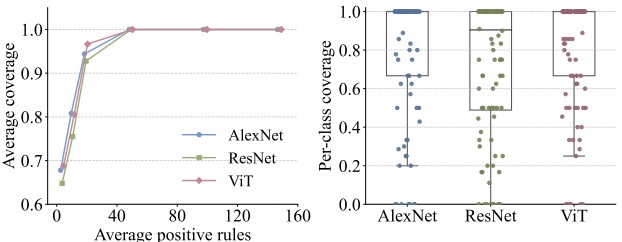

*Figure 4.* Rule sparsity vs. coverage (CIFAR-100). Left: average rule coverage. Right: per-class coverage distribution when retaining only the top 5% positive rules.

on a few head rules instead of many redundant ones. To quantify this, we define rule coverage as, for each backbone $a$ and class $c$, the fraction of correctly classified samples that activate at least one positive rule of class $c$ under $R$; higher coverage means that more correct predictions are explicitly explained by the rules. Figure 4 shows that rule coverage quickly saturates as the number of rules increases, so keeping only about $\sim 20$ positive rules already yields near-perfect coverage and adding more rules brings negligible benefit. The class-wise distributions further show that even with only 5% of the rules, most $(c, a)$ pairs still maintain high coverage, indicating that broadcasting only the pruned positive rules is sufficient and that further expanding the rule set merely introduces redundant patterns.

**Representative rules.** To evaluate rule effectiveness, Table 2 reports representative class-wise rules learned on CIFAR-100, along with their empirical support and balanced accuracy. Support measures how frequently a rule is triggered on its target class, while precision reflects how accurately the rule identifies the target class. We observe that high-weight rules typically achieve both high support and high accuracy, and rules across different classes are largely non-overlapping, suggesting they are stably activated for the target class and highly discriminative, rather than noisy patterns driven by a few incidental samples.

*Table 2.* Class-wise rules learned by the rule network on CIFAR-100. Each rule is a logical combination of threshold predicates, where $s[d_i]$ denotes the $d_i$-th coordinate of input $s$. Weight is the classifier coefficient; only larger-weight rules are shown per class.

| class | weight | rule | support | precision |
|---|---|---|---|---|
| oaktree | 0.81 | $s[19]>0.07 \wedge s[429]>-0.10$ | 0.90 | 0.83 |
| | 0.72 | $s[17]>0.02 \wedge s[122]>-0.10 \wedge s[180]>0.07 \wedge s[507]>0.04$ | 0.82 | 0.74 |
| tiger | 0.85 | $s[125]>-0.05 \wedge s[256]>0.10 \wedge s[349]>0.04$ | 0.83 | 0.82 |
| | 0.73 | $s[5]>0.04 \wedge s[240]>0.04 \wedge s[425]>-0.08$ | 0.86 | 0.90 |

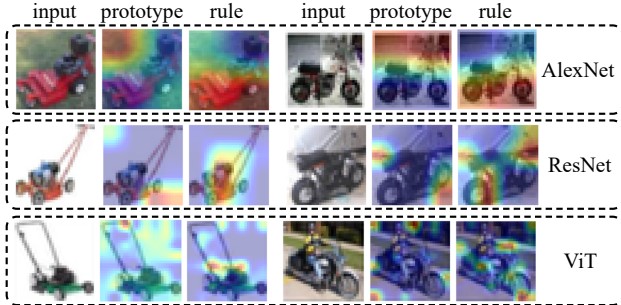

*Figure 5.* Rule interpretability visualization (CIFAR-100). Each column shows one view, input: original image; prototype: CAM heatmap from the response of the similarity between features and mean prototype; rule: CAM heatmap from the response of class rule activations on the image features.

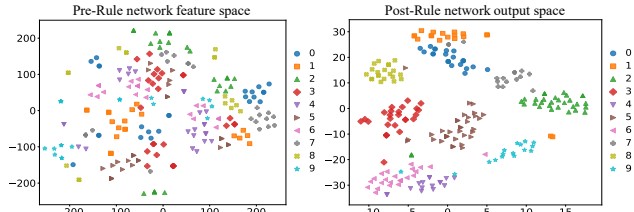

*Figure 6.* Class separability before and after the rule network on CIFAR-10. We apply t-SNE to the same training samples in the pre-rule feature space and in the post-rule output space, using logits as representations in the latter.

**Rule visualization**. We validate the effectiveness of rules via CAM visualizations. Figure 5 shows that rule activations consistently focus on the target object or key discriminative parts, whereas prototype often produces more diffuse responses with noticeable noise. Together with Figure 4, this suggests that rules are not triggered by chance; instead, they capture stable and class-consistent evidence that generalizes across backbones, making them a reliable carrier for cross-architecture collaboration.

**From prototypes to a rule space.** Figure 6 shows that, before entering the rule network, class prototypes are more entangled. After the rule network, classes form tighter clusters with clearer inter-class boundaries. This suggests that the rule network can map heterogeneous features into an architecture-invariant rule space and improve separability. It does so without sharing proxy models or explicitly aggregating features, thereby providing a transferable carrier for collaboration.

### 4.4. Ablation Study

**Loss complementarity.** Figure 7d verifies that the two loss terms are complementary. RFL-rule yields clear gains by injecting sparse, class-relevant logical evidence to stabilize class-conditioned alignment. In contrast, RFL-kd performs significantly worse because the teacher signal is less reliable without an explicit rule-activation constraint. Combining

the two achieves the best accuracy, showing that the rule structure provides a strong anchor while distillation further refines decision boundaries for more effective collaboration.

**Hyperparameter sensitivity.** Figure 7a–7b shows that these hyperparameters are stable in a moderate range and mainly degrade at extremes. $\alpha_{kd}$ and $\alpha_{ru}$ control the strengths of distillation and rule regularization, respectively. Too small yields insufficient signal, while too large may cause over-regularization or amplify early-stage teacher noise. Figure 7c further indicates that a too-small $m$ makes the rule constraint overly weak, whereas a too-large $m$ is hard to satisfy and suppresses local learning. Accordingly, a round-wise increasing weight schedule can reduce early interference and improve robustness.

## 5. Conclusion

This paper revisits heterogeneous federated learning from the perspective of the collaboration carrier, shifting the focus from conventional model/feature alignment to the design of interpretable rule-based knowledge that can be shared across architectures. In strongly heterogeneous scenarios, directly aligning or averaging feature representations produced by different architectures can easily lead to semantic confusion and distorted decision boundaries. To address this issue, we propose RFL, which replaces incompatible feature spaces with an architecture-invariant rule space. By using a lightweight rule network, RFL transforms client-uploaded class prototypes into interpretable class-discriminative rules, enabling collaboration among heterogeneous clients without relying on public data, proxy models, or forced feature

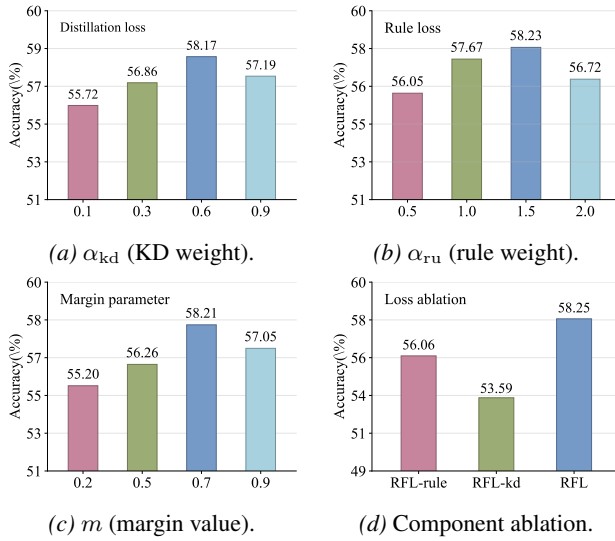

*Figure 7.* Ablation study (CIFAR-100). (a) Sensitivity to the weight $\alpha_{kd}$. (b) Sensitivity to the weight $\alpha_{ru}$. (c) Sensitivity to the margin $m$. (d) Accuracy comparison between RFL and variants without KD (RFL-rule) or rule regularization (RFL-kd).

alignment. Instead, collaboration is reformulated as an interpretable, compact, and selectively absorbable process of rule-based knowledge co-construction.

Experimental results show that, under various data partitions and model-heterogeneous settings, RFL effectively alleviates feature semantic confusion compared with prototype-based methods, and significantly reduces communication and computation overhead compared with proxy-model methods, achieving a better overall trade-off among accuracy, communication cost, and interpretability. Overall, by constructing a rule-based knowledge carrier that is stably expressed across architectures, interpretable, and controllably absorbable, RFL provides a new methodological perspective for federated collaboration under strong heterogeneity. In the future, RFL can be further extended with richer intra-class statistics and stronger rule carriers, enabling it to capture more complex class-discriminative rules and adapt to more challenging vision tasks such as object detection and segmentation.

## Acknowledgements

This work was supported in part by the National Natural Science Foundation of China under Grant U23B2049, Grant 62476194, Grant 62276186, and Grant 22527901; in part by the Fundamental and Interdisciplinary Disciplines Breakthrough Plan of the Ministry of Education of China under Grant JYB2025XDXM503; and in part by the Independent Research Project of the State Key Laboratory of Synthetic Biology under Project HCZC-202606A.

## Impact Statement

This work aims to advance federated learning to address complex heterogeneous scenarios. By exchanging class-level discriminative rule knowledge rather than raw data or complete local models, RFL is expected to provide a more lightweight and interpretable collaboration approach for privacy-conscious and resource-diverse application scenarios. Nevertheless, class-level knowledge summaries are still statistical summaries of local data; therefore, in highly sensitive application domains, additional privacy-preserving mechanisms such as prototype perturbation or differential privacy should still be considered. Overall, RFL does not introduce additional societal risks.

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

# A. Rule Network Details

## A.1. Rule training set construction

**Raw supervised slots.** After local training, client $i$ uploads class prototypes together with its architecture label: $\{p_{i,c}\}_{c \in \mathcal{C}_i}$ and $A_i \in \mathcal{A}$ with $|\mathcal{A}| = A$. The server converts each prototype into an input slot $s_{i,c} = [p_{i,c}; a_i] \in \mathbb{R}^{d+A}$, where $a_i \in \{0, 1\}^A$ is the one-hot architecture code. The raw server dataset is

$$D_{\text{raw}} = \bigcup_{i=1}^{|D_{\text{raw}}|} \{(s_{i,c}, c)\}.$$

**Balanced sub-sampling across architectures (within each class).** For a class $c$ and an architecture label $A' \in \mathcal{A}$, let $\mathcal{P}_{c,A'} = \{[p_{i,c}; a_i] \mid A_i = A'\}$ denote the set of class-$c$ slots contributed by clients whose architecture is $A'$. To avoid a dominant architecture overwhelming rule learning, we sub-sample each $\mathcal{P}_{c,A'}$ to a comparable size. Let $n_{c,A'} = |\mathcal{P}_{c,A'}|$ and $n_c^{\min} = \min_{A': n_{c,A'}>0} n_{c,A'}$. Given a balance ratio $\rho_{\text{bal}} \geq 1$ (default $\rho_{\text{bal}} = 1$), we choose

$$n_{c,A'}^{\text{sub}} = \min\left(n_{c,A'}, \lceil \rho_{\text{bal}} \, n_c^{\min} \rceil\right),$$

and uniformly sample $n_{c,A'}^{\text{sub}}$ prototypes from each $\mathcal{P}_{c,A'}$ without replacement. The resulting set is converted to slots and added to $D_{\text{raw}}$. This keeps the per-class contributions from different backbones comparable while preserving multi-client diversity within each backbone.

**Boundary augmentation within each architecture.** Center prototypes alone may lead to memorizing class centers and weak decision boundaries. we generate a small set of boundary samples within each architecture. For architecture $A'$ and class $c$, let $\mu_{c,A'}$ denote the class mean prototype. We select a few neighboring classes $c'$ by nearest mean distance and synthesize boundary prototypes via light interpolation:

$$p_{\text{bnd}} = \lambda \, \mu_{c,A'} + (1 - \lambda) \, \mu_{c',A'} + \varepsilon, \qquad \lambda \sim \text{Unif}[F - w, F + w],$$

where $F \approx 0.5$ and $w$ controls the boundary width, and $\varepsilon$ is a small Gaussian perturbation. Each boundary sample is paired with the corresponding architecture code to form a slot and added to the server rule training set. The final server dataset is $D_{\text{srv}} = D_{\text{raw}} \cup D_{\text{bnd}}$.

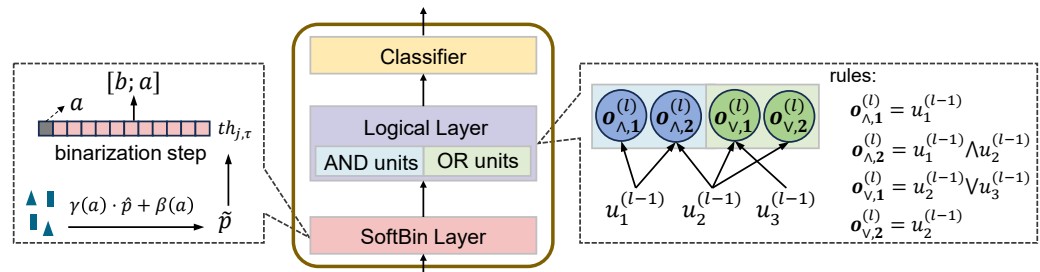

*Figure 8.* Architecture of the rule network. The dashed boxes illustrate the layer-wise configurations, with notation consistent with the main text.

## A.2. Rule network configuration

The rule network maps $s = [p; a]$ to logits $z^R(s)$ via SoftBin and $L$ logic layers, as illustrated in Figure 8.

**Architecture-conditioned normalization and affine calibration.** SoftBin first applies an architecture-conditioned normalization: $\hat{p} = \text{Norm}_a(p)$, followed by an architecture-conditioned affine block:

$$\tilde{p} = \gamma(a) \odot \hat{p} + \beta(a), \qquad \gamma(a) = a^\top \Gamma_a, \; \beta(a) = a^\top \Gamma_b,$$

where $\Gamma_a, \Gamma_b \in \mathbb{R}^{A \times d}$ are trainable parameters and $a \in \{0, 1\}^A$ is one-hot. This calibration aligns the scale and shift of prototype coordinates across backbones, while keeping the subsequent discretization shared.

---

**Algorithm 1** RFL: Server-Client Training via a Rule Network

---

1: **Input:** Clients $\{(D_i)\}_{i=1}^N$; Clients architecture label $A_i$; global rounds $T$; local epochs $E$.
2: **Output:** Personalized client models $\{\theta_i^{(T)}\}_{i=1}^N$ and rule network $\theta_R^{(T)}$.
3: **Initialize.** Each client trains with $\ell_{\text{ce}}$ and uploads $(\{p_{i,c}\}^{(0)}, A_i)$.
4: Server constructs $D_{\text{srv}}^{(0)}$ and trains initial $\theta_R^{(0)}$; extracts $\{\tilde{\mathcal{H}}_c^{+,(0)}\}_{c=1}^C$.
5: **for** $t = 0, 1, \ldots, T-1$ **do**
6:     Server broadcasts $\theta_R^{(t)}$ and $\{\tilde{\mathcal{H}}_c^{+,(t)}\}_{c=1}^C$.
7:     **for all** $i = 1, 2, \ldots, N$ **in parallel do**
8:         **for** $e = 1, 2, \ldots, E$ **do**
9:             Sample a minibatch $(x, y) \sim D_i$ and compute $z_i(x)$ and $e_i(x)$; form $s_i(x) = [e_i(x); a_i]$.
10:             Query frozen rule network: $(z^R(s_i(x)), r(s_i(x))) = R(s_i(x); \theta_R^{(t)})$.
11:             Compute $\ell_{\text{rule}}(s_i(x), y)$ using $\tilde{\mathcal{H}}_y^{+,(t)}$.
12:             Compute $\ell_{\text{kd}}(z^R(s_i(x)), z_i(x))$.
13:             Update $\theta_i$ by minimizing local objective (Eq. (12)).
14:         **end for**
15:         Re-estimate prototypes $\{p_{i,c}\}^{(t+1)}$ and upload.
16:     **end for**
17:     Construct $D_{\text{srv}}^{(t+1)}$.
18:     Train rule network to obtain $\theta_R^{(t+1)}$ and $\{\tilde{\mathcal{H}}_c^{+,(t+1)}\}$.
19: **end for**

---

**Shared thresholds and Boolean atoms.** Each calibrated coordinate $\tilde{p}_j$ is compared to shared thresholds $\{\text{th}_{j,\tau}\}_{\tau=1}^M$:

$$b_{j,\tau}(s) = \mathbb{I}\{\tilde{p}_j(s) > \text{th}_{j,\tau}\}, \qquad b(s) = \text{vec}(\{b_{j,\tau}(s)\}) \in \{0,1\}^{dM}.$$

Because thresholds are shared across architectures, atoms $b_{j,\tau}$ live in a common discrete domain for all backbones. (In practice, thresholds can be fixed after initialization for stability, or made learnable by treating them as trainable parameters.)

### A.3. Depth configuration: why one layer is usually sufficient

In our setting, each input is a class-level summary $p_{i,c}$ rather than a raw instance feature. This already reduces intra-class variance and concentrates discriminative evidence into a compact representation. Empirically, a single logic layer ($L = 1$) typically provides enough expressive power to compose stable rule evidence while keeping optimization stable and rules interpretable. We only consider $L = 2$ for substantially more complex tasks (e.g., larger label space), where deeper compositions may capture higher-order interactions.

### A.4. Why this is cross-architecture friendly.

Cross-architecture fusion happens *after* discretization: (i) SoftBin calibrates backbone-specific scales via $(\gamma(a), \beta(a))$; (ii) atoms $b_{j,m}$ are defined by shared thresholds and hence comparable across backbones; (iii) logic layers apply architecture-agnostic connection matrices to $[b; a]$, embedding heterogeneous prototypes into a shared rule space. Therefore, prototypes from different architectures can be pooled to learn a unified rule library without averaging incompatible feature geometries.

## B. Overall Algorithm

The complete training procedure of our framework is provided in Algorithm 1.

## C. Proofs for Section 3.3

### C.1. Architecture-invariant rule space

We first give a constructive justification for Definition 3.1. For an input slot $s = [p; a]$, the SoftBin module in Eq. (4) first normalizes and calibrates the prototype $p$ via an architecture-conditioned affine transformation, and then compares each

calibrated dimension $\tilde{p}_j$ with a shared set of thresholds $\{\text{th}_{j,\tau}\}_{\tau=1}^M$:

$$b_{j,\tau}(s) = \mathbb{I}\{\tilde{p}_j - \text{th}_{j,\tau} > 0\}, \qquad j = 1, \ldots, d, \ \tau = 1, \ldots, M.$$

All Boolean atoms are flattened into $b(s) = \text{vec}(\{b_{j,m}(s)\}) \in \{0,1\}^{dM}$. The architecture code $a$ is concatenated back and fed into the logic layers, which apply the same discrete connection matrices to $[b(s); a]$ for all backbones according to Eqs. (5)–(6). After $L$ logic layers we obtain the final rule activation vector $r(s) = u^{(L)}(s) \in [0,1]^{d_L}$, and we define the rule representation to be

$$\Pi(s) := r(s).$$

By construction, the mapping $s \mapsto \Pi(s)$ is implemented by the *same* composition of SoftBin and logic operations for all backbones; the only architecture-dependent part is the affine calibration $(\gamma(a), \beta(a))$ before discretization. Once the Boolean atoms $b_{j,m}$ are formed, they live in a common discrete domain and are directly comparable across architectures. The logic layers then operate on $[b(s); a]$ via architecture-agnostic connection matrices, so $\Pi(s)$ lies in a shared rule space $\mathcal{R} = \Pi(S)$ independent of which backbone produced the original prototype.

The rule hypothesis class in Definition 3.1 is

$$H_R = \left\{ \varphi(r) = W_{\text{cls}} r \mid W_{\text{cls}} \in \mathbb{R}^{C \times d_L} \right\},$$

Our rule network $R$ has the form $R(s) = \varphi(\Pi(s))$ for some $\varphi \in \mathcal{H}_R$, with final rule logits $z_R(s) = W_{\text{cls}} \Pi(s)$.

**Lemma C.1.** *For any two slots $s, s' \in S$, if $\Pi(s) = \Pi(s')$, then for all $\varphi \in \mathcal{H}_R$,*

$$z_R(s) = \varphi(\Pi(s)) = \varphi(\Pi(s')) = z_R(s').$$

*Proof.* Let $\varphi(r) = W_{\text{cls}} r$ be any element in $\mathcal{H}_R$. If $\Pi(s) = \Pi(s')$, then

$$z_R(s) = W_{\text{cls}} \Pi(s) = W_{\text{cls}} \Pi(s') = z_R(s').$$

Therefore the prediction of $R$ depends only on the rule representation $\Pi(s)$ and is invariant to which backbone produced the original prototype. This is exactly the invariance claimed in Definition 3.1. $\square$

In this sense, heterogeneous backbones are embedded into a common rule space $\mathcal{R} = \Pi(S)$ and equivalently reparameterized by a single shared classifier in $\mathcal{H}_R$, which is the basis for analyzing collaboration in the rule space rather than in the original feature spaces.

### C.2. Selective absorption in the rule space

For a sample with label $c$, let $\tilde{H}_c^+ \subseteq \{1, \ldots, d_L\}$ be the pruned positive-rule set for class $c$ and $K_c^+ = |\tilde{H}_c^+|$. The rule loss in Eq. (10) is

$$\ell_{\text{rule}}(s, y = c) = \frac{1}{|\tilde{H}_c^+|} \sum_{k \in \tilde{H}_c^+} \left[ \max\left(0, \, m - r_k(s)\right) \right]^2,$$

where $m \in (0,1)$ is a fixed margin and $r_k(s)$ is the activation of the $k$-th logic unit.

For any margin buffer $\eta \in (0, m)$ and slot $s$ with label $c$, define the set of severely under-activated positive rules

$$V_c(s, \eta) = \left\{ k \in \tilde{H}_c^+ \mid r_k(s) \leq m - \eta \right\}.$$

Lemma 3.2 claims that the rule loss controls both the fraction of severely violated positive rules and the total activation of the positive-rule set:

$$\frac{|V_c(s, \eta)|}{|\tilde{H}_c^+|} \leq \frac{\ell_{\text{rule}}(s, y = c)}{\eta^2}, \quad \sum_{k \in \tilde{H}_c^+} r_k(s) \geq |\tilde{H}_c^+|(m - \eta)\left(1 - \frac{\ell_{\text{rule}}(s, y = c)}{\eta^2}\right).$$

*Proof of Lemma 3.2.* Fix a label $c$ and a slot $s$.

**(i) Fraction of severely violated positive rules.** By definition of $V_c(s, \eta)$, for any $k \in V_c(s, \eta)$ we have

$$r_k(s) \leq m - \eta \quad \Longrightarrow \quad m - r_k(s) \geq \eta,$$

and hence

$$\left[\max(0, m - r_k(s))\right]^2 \geq \eta^2.$$

Therefore

$$\ell_{\mathrm{rule}}(s, y = c) = \frac{1}{|\tilde{H}_c^+|} \sum_{k \in \tilde{H}_c^+} \left[\max(0, m - r_k(s))\right]^2$$

$$\geq \frac{1}{|\tilde{H}_c^+|} \sum_{k \in V_c(s, \eta)} \left[\max(0, m - r_k(s))\right]^2$$

$$\geq \frac{1}{|\tilde{H}_c^+|} \sum_{k \in V_c(s, \eta)} \eta^2 = \frac{|V_c(s, \eta)|}{|\tilde{H}_c^+|} \eta^2.$$

Rearranging yields

$$\frac{|V_c(s, \eta)|}{|\tilde{H}_c^+|} \leq \frac{\ell_{\mathrm{rule}}(s, y = c)}{\eta^2},$$

which proves the first inequality.

**(ii) Lower bound on the total activation.** Let $J = V_c(s, \eta)$. For any $k \in \tilde{H}_c^+ \setminus J$ we have $r_k(s) > m - \eta$, hence

$$\sum_{k \in \tilde{H}_c^+} r_k(s) = \sum_{k \in \tilde{H}_c^+ \setminus J} r_k(s) + \sum_{k \in J} r_k(s) \geq \sum_{k \in \tilde{H}_c^+ \setminus J} (m - \eta) = (K_c^+ - |J|)(m - \eta).$$

From part (i) we have $|J| \leq K_c^+ \ell_{\mathrm{rule}}(s, y = c)/\eta^2$, so

$$\sum_{k \in \tilde{H}_c^+} r_k(s) \geq \left(K_c^+ - K_c^+ \frac{\ell_{\mathrm{rule}}(s, y = c)}{\eta^2}\right)(m - \eta)$$

$$= K_c^+ (m - \eta)\left(1 - \frac{\ell_{\mathrm{rule}}(s, y = c)}{\eta^2}\right),$$

which is the second inequality.

In particular, if $\ell_{\mathrm{rule}}(s, y = c) \leq \Lambda$ with $\Lambda \ll \eta^2$, then at most a small fraction of positive rules can stay far below $m - \eta$, and most class-positive rules are sufficiently activated in the shared rule space. $\square$

### C.3. Rule-space margin lower bound

We now show how the rule constraint induces a margin lower bound in the rule space, as formalized in Lemma 3.4. From Assumption 3.3, the rule activation vector satisfies $r(s) \in [0, 1]^{d_L}$ and the rule logits of $R$ are $z_c^R(s) = \sum_{k=1}^{d_L} w_{c,k} r_k(s)$, where $w_c \in \mathbb{R}^{d_L}$ is the classifier weight vector for class $c$. For each class $c$ and each index $k \in \{1, \ldots, d_L\}$, if $k \in \tilde{\mathcal{H}}_c^+$ then $w_{c,k} - \max_{c' \neq c} w_{c',k} \geq \alpha_{\mathrm{pos}} > 0$, and we also have $\|w_c\|_2 \leq B_w$.

For any labeled slot $(s, y)$, the rule-space margin is defined as

$$\mathrm{margin}(s, y) = z_y^R(s) - \max_{c' \neq y} z_{c'}^R(s).$$

Lemma 3.4 states that if the rule loss is small, then the rule-space margin admits a positive lower bound that depends explicitly on the rule parameters. Formally, for each class $c$ let $K_c^+ = |\tilde{\mathcal{H}}_c^+|$ and $K_{\min}^+ = \min_c K_c^+$. Under Assumption 3.3 and $r_k(s) \in [0, 1]$, for any $\eta \in (0, m)$ and any sample $(s, y = c)$ satisfying $\ell_{\mathrm{rule}}(s, y = c) \leq \Lambda$, the rule-space margin satisfies

$$\mathrm{margin}(s, y) \geq \Gamma(\Lambda, \eta),$$

where

$$\Gamma(\Lambda, \eta) = \alpha_{\mathrm{pos}} K_{\min}^+ (m - \eta)\left(1 - \frac{\Lambda}{\eta^2}\right) - 2B_w \sqrt{d_L}.$$

*Proof of Lemma 3.4.* Fix a labeled slot $(s, y = c)$ and a buffer $\eta \in (0, m)$. For any competing class $c' \neq c$, we can write the rule-logit difference as

$$z_c^R(s) - z_{c'}^R(s) = \sum_{k=1}^{d_L} (w_{c,k} - w_{c',k}) \, r_k(s).$$

We decompose the sum into the contribution from the selected positive-rule set $\tilde{\mathcal{H}}_c^+$ and that from the remaining rules:

$$z_c^R(s) - z_{c'}^R(s) = \underbrace{\sum_{k \in \tilde{\mathcal{H}}_c^+} (w_{c,k} - w_{c',k}) \, r_k(s)}_{\text{contribution of class-}c\text{ positive rules}} + \underbrace{\sum_{k \notin \tilde{\mathcal{H}}_c^+} (w_{c,k} - w_{c',k}) \, r_k(s)}_{\text{contribution of remaining rules}}.$$

**(i) Contribution of positive rules.** By Assumption 3.3(i), for any $k \in \tilde{\mathcal{H}}_c^+$ we have

$$w_{c,k} - w_{c',k} \geq \alpha_{\text{pos}}, \quad \forall c' \neq c.$$

Hence

$$\sum_{k \in \tilde{\mathcal{H}}_c^+} (w_{c,k} - w_{c',k}) \, r_k(s) \geq \alpha_{\text{pos}} \sum_{k \in \tilde{\mathcal{H}}_c^+} r_k(s).$$

Applying Lemma 3.2 with label $c$ and $\ell_{\text{rule}}(s, y = c) \leq \Lambda$ yields

$$\sum_{k \in \tilde{\mathcal{H}}_c^+} r_k(s) \geq |\tilde{\mathcal{H}}_c^+|(m - \eta)\left(1 - \frac{\Lambda}{\eta^2}\right) \geq K_{\min}^+(m - \eta)\left(1 - \frac{\Lambda}{\eta^2}\right),$$

where $K_{\min}^+ = \min_c |\tilde{\mathcal{H}}_c^+|$. Therefore,

$$\sum_{k \in \tilde{\mathcal{H}}_c^+} (w_{c,k} - w_{c',k}) \, r_k(s) \geq \alpha_{\text{pos}} K_{\min}^+ (m - \eta)\left(1 - \frac{\Lambda}{\eta^2}\right). \tag{14}$$

**(ii) Contribution of remaining rules.** For the sum over $k \notin \tilde{\mathcal{H}}_c^+$, we use a norm-based upper bound. Since $r_k(s) \in [0, 1]$ for all $k$, we have $\|r(s)\|_2 \leq \sqrt{d_L}$. By the triangle inequality and Cauchy–Schwarz,

$$\left| \sum_{k \notin \tilde{\mathcal{H}}_c^+} (w_{c,k} - w_{c',k}) \, r_k(s) \right| \leq \sum_{k=1}^{d_L} |w_{c,k} - w_{c',k}| \, r_k(s) \leq \|w_c - w_{c'}\|_2 \, \|r(s)\|_2.$$

Assumption 3.3(ii) implies $\|w_c - w_{c'}\|_2 \leq \|w_c\|_2 + \|w_{c'}\|_2 \leq 2B_w$, hence

$$\sum_{k \notin \tilde{\mathcal{H}}_c^+} (w_{c,k} - w_{c',k}) \, r_k(s) \geq -2B_w \sqrt{d_L}. \tag{15}$$

**(iii) Margin lower bound.** Combining (14) and (15), we obtain that for any $c' \neq c$,

$$z_c^R(s) - z_{c'}^R(s) \geq \alpha_{\text{pos}} K_{\min}^+ (m - \eta)\left(1 - \frac{\Lambda}{\eta^2}\right) - 2B_w \sqrt{d_L}.$$

Taking the minimum over all $c' \neq c$ yields

$$\text{margin}(s, y) = \min_{c' \neq c}\left(z_c^R(s) - z_{c'}^R(s)\right) \geq \alpha_{\text{pos}} K_{\min}^+ (m - \eta)\left(1 - \frac{\Lambda}{\eta^2}\right) - 2B_w \sqrt{d_L} = \Gamma(\Lambda, \eta),$$

which completes the proof. □

**Discussion.** Lemma 3.4 makes explicit how the rule constraint $\ell_{\mathrm{rule}}$ shapes the margin in the shared rule space:

- The term $\alpha_{\mathrm{pos}} K_{\min}^+ (m - \eta)$ reflects the *strength and redundancy* of positive rules: more reliable positive rules and a larger fixed margin $m$ increase the achievable rule-space margin.

- The factor $(1 - \Lambda/\eta^2)$ quantifies the effect of the rule loss: driving $\ell_{\mathrm{rule}}$ below a small $\Lambda$ forces the majority of positive rules in $\tilde{\mathcal{H}}_c^+$ to be activated close to $m$, thereby increasing their aggregate contribution to the true class relative to competing classes.

- The term $2B_w \sqrt{d_L}$ captures the worst-case fluctuation introduced by the remaining rules and inter-class weight differences under the weight-norm constraint, and can be viewed as a complexity penalty of the rule classifier.

Together with standard margin-based generalization bounds (Bartlett & Mendelson, 2003), Lemma 3.4 shows that rule-regularized local learning not only provides a controllable mechanism for selective absorption in the rule space, but also shrinks the effective hypothesis space by enforcing a positive rule-space margin under bounded classifier norm.

### C.4. Rule-regularized local generalization

In RFL, the local training objective on client $i$ in round $t$ takes the form

$$\mathcal{L}_i(\theta_i) = \underbrace{\mathbb{E}_{(x,y)\sim\mathcal{D}_i}\left[\ell_{\mathrm{ce}}(f_i(x;\theta_i), y)\right]}_{\text{CE term}} + \underbrace{\alpha_{kd}\,\mathcal{L}_i^{\mathrm{kd}}(\theta_i)}_{\text{KD term}} + \underbrace{\alpha_{ru}\,\mathcal{L}_i^{\mathrm{rule}}(\theta_i)}_{\text{rule term}}. \tag{16}$$

Here the CE term is the standard classification loss of the local model $f_i$ on $\mathcal{D}_i$, the KD term encourages $f_i$ to align its logits with those of the frozen rule network, and the rule term

$$\mathcal{L}_i^{\mathrm{rule}}(\theta_i) = \mathbb{E}_{(x,y)\sim\mathcal{D}_i}\left[\ell_{\mathrm{rule}}(s_i(x), y)\right]$$

penalizes insufficient activation of class-consistent positive rules in the shared rule space. Theorem 3.5 concerns the *0–1 risk* of the final local model $f_i$, and the proof proceeds by analyzing how these three components jointly affect the generalization behavior.

*Proof of Theorem 3.5.* For clarity we decompose the argument into three steps.

**(i) Local risk decomposition and alignment error.** Since $s_i(x)$ denotes the slot representation of an input $x$ on client $i$, let $z_c^R(s_i(x))$ be the logit of the frozen rule network for class $c$, and define its prediction as

$$\hat{y}^R(s_i(x)) := \arg\max_c z_c^R(s_i(x)).$$

Similarly, let $z_c^i(x)$ be the logit of the local classifier $f_i$ for class $c$, and define its prediction by

$$\hat{y}_i(x) := \arg\max_c z_c^i(x).$$

Then for each $(x, y)$, we have the pointwise decomposition

$$\mathbb{I}\{\hat{y}_i(x) \neq y\} \;\leq\; \mathbb{I}\{\hat{y}^R(s_i(x)) \neq y\} + \mathbb{I}\{\hat{y}_i(x) \neq \hat{y}^R(s_i(x))\}, \tag{17}$$

where the event $\hat{y}_i(x) \neq y$ is contained in the union of (i) the rule network misclassifying $(s_i(x), y)$ and (ii) the local model disagreeing with the rule network on $x$.

Taking expectation over $(x, y) \sim \mathcal{D}_i$ yields

$$L_i(f_i) := \mathbb{E}_{(x,y)\sim\mathcal{D}_i}\left[\mathbb{I}\{\hat{y}_i(x) \neq y\}\right] \;\leq\; L_i^{\mathrm{rule}} + \varepsilon_i^{\mathrm{align}}, \tag{18}$$

where $L_i^{\mathrm{rule}} := \mathbb{E}_{(x,y)\sim\mathcal{D}_i}\left[\mathbb{I}\{\hat{y}^R(s_i(x)) \neq y\}\right], \varepsilon_i^{\mathrm{align}} := \Pr_{x\sim\mathcal{D}_i}\left(\hat{y}_i(x) \neq \hat{y}^R(s_i(x))\right).$

The KD term in (16) is precisely what drives the local logits $z_c^i(x)$ towards the rule logits $z_c^R(s_i(x))$ during training. In Theorem 3.5 we do not bound $\varepsilon_i^{\mathrm{align}}$ explicitly in terms of $\alpha_{kd}$, but instead treat it as an explicit alignment error term that can be estimated or controlled in practice. Therefore, to prove the theorem it suffices to upper bound the *rule-space risk* $L_i^{\mathrm{rule}}$ and then use (18).

**(ii) From rule loss to a rule-space margin lower bound.** Let $D_i = \{(x_j, y_j)\}_{j=1}^{n_i}$ be the local dataset on client $i$, drawn i.i.d. from $\mathcal{D}_i$, and denote by $s_j := s_i(x_j)$ the corresponding slots. By the definition of $\Lambda_i$ in Theorem 3.5, we assume that after local training the rule loss on all training samples is bounded as

$$\ell_{\mathrm{rule}}(s_j, y_j) \leq \Lambda_i \quad \text{for all } j = 1, \dots, n_i.$$

Under Assumption 3.3 on the rule weights and $r_k(s) \in [0, 1]$, Lemma 3.4 shows that the rule loss bound implies a uniform rule-space margin lower bound. More precisely, for any fixed $\eta \in (0, m)$ we have

$$\mathrm{margin}(s_j, y_j) \geq \Gamma(\Lambda_i, \eta) := \alpha_{\mathrm{pos}} K_{\mathrm{min}}^+ (m - \eta)\left(1 - \frac{\Lambda_i}{\eta^2}\right) - 2B_w\sqrt{d_L}, \tag{19}$$

for all training samples $(s_j, y_j)$. $K_{\mathrm{min}}^+$ is the minimal number of positive rules across classes, $d_L$ is the dimensionality of the rule representation, and $\alpha_{\mathrm{pos}}, B_w$ are the constants from Assumption 3.3.

Intuitively, the rule term (c) in (16) enforces a small rule loss $\Lambda_i$, which forces the majority of class-consistent positive rules to be strongly activated on $(s_j, y_j)$ and guarantees a uniform margin lower bound $\Gamma(\Lambda_i, \eta)$ in the shared rule space via Lemma 3.4.

For any margin parameter $\kappa > 0$ we define the empirical margin error of the rule classifier on $D_i$ as

$$\hat{L}_i^\kappa(\phi) := \frac{1}{n_i} \sum_{j=1}^{n_i} \mathbb{I}\{\mathrm{margin}(s_j, y_j) \leq \kappa\}.$$

By (19), whenever $0 < \kappa < \Gamma(\Lambda_i, \eta)$ we obtain

$$\hat{L}_i^\kappa(\phi) = 0. \tag{20}$$

**(iii) Margin-based generalization in the shared rule space.** We now apply a standard margin-based generalization bound to the shared rule classifier $\varphi$ used in the rule network $R$. The rule representation $r(s)$ lies in $[0, 1]^{d_L}$, and the class weight vectors $\{w_c\}_{c=1}^C$ of $\varphi$ satisfy $\|w_c\|_2 \leq B_w$ for all $c$. $H_R$ denotes the linear multi-class rule hypothesis class in the rule space, so that $\varphi \in H_R$.

The CE term in (16) encourages small training error of $f_i$ on $D_i$, and via the KD alignment it also encourages $\varphi$ to predict correctly on those samples. In the proof we do not need to quantify this effect separately. We will use the margin lower bound (19), which already encodes the fact that (on $D_i$) the rule logits $z_R(s_j)$ separate the true class from the others by at least $\Gamma(\Lambda_i, \eta)$.

The hypothesis class $H_R$ has empirical Rademacher complexity bounded as

$$\mathfrak{R}_{n_i}(H_R) \leq C_1 \frac{B_w\sqrt{d_L}}{\sqrt{n_i}}$$

for some universal constant $C_1 > 0$. This follows from the standard bound for linear predictors on $\ell_2$-bounded weights and $\ell_2$-bounded features, where $\|r(s_j)\|_2 \leq \sqrt{d_L}$ holds for all $j$. Then a classical margin-based generalization bound (Bartlett & Mendelson, 2003) implies that, for any $\kappa > 0$ and any $\delta \in (0, 1)$, with probability at least $1 - \delta$ over $D_i \sim \mathcal{D}_i$, we have

$$L_i^{\mathrm{rule}} \leq \hat{L}_i^\kappa(\varphi) + \frac{C_2 B_w\sqrt{d_L}}{\kappa\sqrt{n_i}} + \sqrt{\frac{\log(1/\delta)}{2n_i}}, \tag{21}$$

for some universal constant $C_2 > 0$.

Combining (21) with (20) and choosing

$$\kappa = \frac{1}{2}\Gamma(\Lambda_i, \eta)$$

gives

$$L_i^{\mathrm{rule}} \leq \frac{2C_2 B_w\sqrt{d_L}}{\Gamma(\Lambda_i, \eta)\sqrt{n_i}} + \sqrt{\frac{\log(1/\delta)}{2n_i}}. \tag{22}$$

Letting $C_1 := 2C_2$ yields exactly

$$L_i^{\text{rule}} \leq C_1 \frac{B_w \sqrt{d_L}}{\Gamma(\Lambda_i, \eta) \sqrt{n_i}} + \sqrt{\frac{\log(1/\delta)}{2n_i}}.$$

Finally, combining the rule-space bound (22) with the risk decomposition (18), we obtain, with probability at least $1 - \delta$,

$$L_i(f_i) \leq C_1 \frac{B_w \sqrt{d_L}}{\Gamma(\Lambda_i, \eta) \sqrt{n_i}} + \varepsilon_i^{\text{align}} + \sqrt{\frac{\log(1/\delta)}{2n_i}}.$$

This is precisely the local generalization bound stated in Theorem 3.5.

$\square$

## D. Additional Experimental Details

### D.1. Experimental Setup

**Datasets and Preprocessing** We evaluate on three standard image classification benchmarks: CIFAR-10, CIFAR-100, and Tiny-ImageNet. For CIFAR-10/100, images are of size $32 \times 32$; for Tiny-ImageNet, images are of size $64 \times 64$. We apply standard, lightweight data augmentation on the training split and use deterministic preprocessing for validation and testing. All images are normalized channel-wise using dataset-specific mean and standard deviation.

To characterize statistical heterogeneity, we consider two widely used partition protocols: **Pathological (Pat)** and **Dirichlet (Dir)**. Following the classical FedAvg-style pathological split, each client is assigned samples from a small subset of classes, yielding severe label skew. Specifically, each client contains 3 classes on CIFAR-10, 15 classes on CIFAR-100, and 30 classes on Tiny-ImageNet. We additionally adopt the practical Dirichlet split in which, for each class, the proportions of its samples allocated to clients are drawn from a Dirichlet distribution with concentration parameter $\alpha$.

**Model Heterogeneity** We study model heterogeneity by assigning different backbone architectures to clients. To evaluate the effectiveness of our method under stronger model heterogeneity, we do not restrict ourselves to models from the same family or sharing a common backbone, but instead adopt a more diverse set of architectures, resulting in a substantially more challenging setting. The heterogeneous backbone sets are: (i) CIFAR-10: AlexNet / MobileNet / ResNet10; (ii) CIFAR-100: AlexNet / ResNet10 / ViT-B/16; (iii) Tiny-ImageNet: MobileNetV3 / ResNet18 / ViT-B/16. Across all settings, the client population follows a fixed architecture ratio of $3:4:3$ over the three backbones.

### D.2. Baselines

We compare with representative heterogeneous federated learning baselines, including FedProto, FedGen, FedTGP, FedMRL, FedZGE, FedSA, and FedSCE. For clarity, we group them by the type of knowledge-sharing carrier: prototype-based methods (e.g., FedProto, FedTGP, FedSA) and proxy-model / distillation-based methods (e.g., FedGen, FedZGE, FedMRL, FedSCE). We adopt a unified fairness protocol by keeping the data partitions, heterogeneous backbones, local training budget, and evaluation procedure identical across methods.

**prototype-based methods.** FedProto, FedTGP, and FedSA use class prototypes as the knowledge-sharing carrier among heterogeneous clients, without introducing any auxiliary models or proxy data. Our method likewise derives discriminative rules from class prototypes, ensuring a consistent source of collaborative knowledge and requiring no additional proxy data.

**proxy-model / distillation-based methods.** FedMRL, and FedSCE are proxy-model-based baselines, where the proxy network is trained and updated on each client's local data. Following the original setups to keep the proxy lightweight, we use a small CNN with two convolutional layers and three fully connected layers. FedGen, in contrast, trains a server-side generator to synthesize pseudo-features for client training; in our implementation, the generator is a simple multilayer perceptron (MLP). FedZGE uses server-generated synthetic data as the collaboration carrier. In each round, the server queries client models with synthetic samples, clients return only the prediction logits on these samples, and the server updates its generator and global model via zeroth-order gradient estimation, without exchanging local model parameters.

*Table 3.* Theoretical communication cost per global round (number of scalars transmitted between the server and all participating clients).

| Method | Communication cost per round |
|---|---|
| FedProto | $\sum_{i=1}^{N} K\,(C_i + C)$ |
| FedTGP | $\sum_{i=1}^{N} K\,(C_i + C)$ |
| FedSA | $\sum_{i=1}^{N} K\,C_i \; + \; NKC$ |
| FedGen | $\sum_{i=1}^{N}\big(2\,|\theta_i^h| + |\theta_{gen}|\big)$ |
| FedMRL | $2N|\theta_{prox}|$ |
| FedZGE | $N|\hat{X}|\big((q+1)d_x + (q+2)C\big)$ |
| FedSCE | $2N|\theta_{prox}|$ |
| RFL (ours) | $K\sum_{i=1}^{N} C_i \; + \; N\big(|\theta_R| + H\big)$ |

## D.3. Communication cost.

We measure the communication cost as the total number of scalar values exchanged between the server and all participating clients in one global round. Throughout this part we use the following notation:

- $N$: number of participating clients per round.

- $C$: total number of classes in the label space.

- $C_i$: number of classes that actually appear on client $i$.

- $K$: dimension of class prototypes / semantic anchors (e.g., $K{=}512$ in our experiments).

- $|\theta_i^h|$: number of parameters in the local classifier on client $i$.

- $|\theta_{prox}|$: number of parameters in the homogeneous proxy model.

- $|\theta_{gen}|$: number of parameters in the generator of FedGen.

- $|\theta_R|$: number of parameters of the server-side rule network $R$ in our method.

- $H$: total number of transmitted positive rule indices across all classes in our method, $H \triangleq \sum_{c=1}^{C} |\tilde{H}_c^+|$, where $\tilde{H}_c^+$ is the pruned positive-rule set for class $c$.

Unless otherwise stated, we only count the dominant terms and ignore negligible constants. We follow the communication-cost computation protocol in FedTGP (Zhang et al., 2024b). For methods already covered in FedTGP, we omit further discussion. For the newly introduced methods, the communication cost is computed as follows:

**FedMRL.** FedMRL also maintains a homogeneous model shared by all clients. In each global round, the server broadcasts $\theta_{\mathrm{prox}}$ to all $N$ participating clients, and each client jointly trains this small model together with its local heterogeneous model and personalized projector. After local training, client $i$ uploads only the updated small-model parameters $\theta_{\mathrm{prox}}$ to the server for aggregation. Therefore, for each client we transmit $|\theta_{\mathrm{prox}}|$ scalars on the downlink and $|\theta_{\mathrm{prox}}|$ on the uplink, and the total communication cost per round is

$$\mathrm{Comm}_{\mathrm{FedMRL}} \;=\; 2N\,|\theta_{\mathrm{prox}}|.$$

**FedSA.** FedSA follows the prototype-based protocol. In each round, client $i$ uploads its local class prototypes $\{p_{i,c} \in \mathbb{R}^K : c \in \mathcal{C}_i\}$ to the server, whose uplink cost is $KC_i$ scalars. The server aggregates these to global prototypes and updates

class-wise semantic anchors (each in $\mathbb{R}^K$), and then broadcasts a set of $C$ anchors to every client, incurring $MKC$ scalars on the downlink. Therefore the total cost per round is

$$\text{Comm}_{\text{FedSA}} = \sum_{i=1}^{N} KC_i + NKC.$$

**FedZGE.** FedZGE performs data-free black-box queries on client models using server-synthesized inputs. In each round, the server samples a synthesized set $\hat{X}$ of size $|\hat{X}|$ from its generator and sends it to every participating client, where each input has dimensionality $d_x$ (e.g., $d_x = 3HW$ for an $H \times W$ image). To estimate zeroth-order gradients, the server additionally constructs $q$ perturbed copies of the same set and distributes them as well (we fix $q = 10$ in all experiments), so each client receives $(q+1)$ input sets in total. Each client returns the predicted logits (of dimension $C$ per sample) on all queried inputs, and the server further broadcasts one set of ensembled logits on $\hat{X}$ back to the clients for local distillation. Therefore, the downlink cost per round is $N|\hat{X}|\big((q+1)d_x + C\big)$ and the uplink cost is $N|\hat{X}|(q+1)C$. Summing both directions yields

$$\text{Comm}_{\text{FedZGE}} = N|\hat{X}|\big((q+1)d_x + (q+2)C\big).$$

**FedSCE.** FedSCE maintains a homogeneous auxiliary model with parameters $\theta_{prox}$ on the server. In each round, the server broadcasts $\theta_{prox}$ to all clients. Client $i$ locally updates the auxiliary model to $\theta_{prox,i}$ and additionally computes two scalar distances $F_{i,1}$ and $F_{i,2}$ that describe its update bias in feature and parameter spaces; then it uploads $(\theta_{prox,i}, F_{i,1}, F_{i,2})$ to the server. Since $F_{i,1}$ and $F_{i,2}$ are distance values, their computational cost is negligible and can be ignored in the overhead analysis. Hence, for each client we have $|\theta_{prox}|$ scalars on the downlink and $|\theta_{prox}|$ on the uplink, and the total cost is

$$\text{Comm}_{\text{FedSCE}} = N|\theta_{prox}| + N|\theta_{prox}| = 2N|\theta_{prox}|.$$

**RFL.** In RFL, the server never aggregates backbone parameters. Instead, collaboration is mediated purely through class prototypes and a shared rule network $R$.

*Client-to-server (uplink).* At the end of the local stage, client $i$ uploads the tuple $(\{p_{i,c}\}, A_i)$ to the server. The dominant cost comes from $\{p_{i,c}\}$ and equals $KC_i$ scalars; the architecture code $A_i$ is a short one-hot vector and is negligible compared to $KC_i$. Summing over all clients gives the uplink cost

$$\text{Comm}_{\text{up}}^{\text{RFL}} = K\sum_{i=1}^{N} C_i.$$

*Server-to-client (downlink).* The server aggregates all uploaded prototypes and trains the rule network $R$ with parameters $\theta_R$. After convergence, it extracts for each class $c$ a pruned set of positive rules $\tilde{H}_c^+$ (indices of rule units) and broadcasts the pair $(\theta_R, \{\tilde{H}_c^+\}_{c=1}^C)$ to all $M$ clients. Let

$$H \triangleq \sum_{c=1}^{C} |\tilde{H}_c^+|$$

be the total number of transmitted rule indices. Then each client receives $|\theta_R| + H$ scalars, and the total downlink cost is

$$\text{Comm}_{\text{down}}^{\text{RFL}} = N\big(|\theta_R| + H\big).$$

*Total cost.* Combining both directions, the overall communication cost per round of RFL is

$$\text{Comm}_{\text{RFL}} = \text{Comm}_{\text{up}}^{\text{RFL}} + \text{Comm}_{\text{down}}^{\text{RFL}} = K\sum_{i=1}^{N} C_i + N\big(|\theta_R| + H\big),$$

as reported in the last row of Table 3. The communication overhead measured in the experiments is reported in Table 1 of the main text.

### D.4. Additional analysis of rule interpretability

**Rule coverage (Figure 4).** For each backbone $a$ and class $c$, let $D_{c,a}^{\mathrm{corr}}$ and the pruned positive-rule set $H_{c,a}^+$ be as in the main text. We say that a sample $x \in D_{c,a}^{\mathrm{corr}}$ is *covered* if at least one positive rule in $H_{c,a}^+$ fires on its slot representation, i.e., there exists $r \in H_{c,a}^+$ with $r(x) = 1$. The rule coverage for the pair $(c, a)$ is then

$$\mathrm{Cov}(c, a) \;=\; \frac{1}{|D_{c,a}^{\mathrm{corr}}|} \sum_{x \in D_{c,a}^{\mathrm{corr}}} \mathbb{I}\big[\exists\, r \in H_{c,a}^+ \text{ s.t. } r(x) = 1\big], \tag{23}$$

where $\mathbb{I}[\cdot]$ is the indicator function. The left panel of Figure 4 reports the average of $\mathrm{Cov}(c, a)$ over all classes and backbones, plotted against the average number of retained positive rules per $(c, a)$. To obtain the sparsity–coverage curve, we sort the positive rules in $H_{c,a}^+$ by their support (defined below) and, for each truncation level $m$, recompute (23) using only the top-$m$ rules. The right panel of Figure 4 shows the distribution of $\mathrm{Cov}(c, a)$ when only the top $5\%$ positive rules are kept for each $(c, a)$.

**Support and balanced precision (Table 2).** For a rule $g$ and a class $c$, we write

$$\mathbb{P}\big(g(x) = 1 \mid y(x) = c\big) \quad \text{and} \quad \mathbb{P}\big(g(x) = 1 \mid y(x) \neq c\big)$$

for the conditional activation probabilities of $g$ on class-$c$ samples and on non-$c$ samples, respectively. The *support* of $g$ for class $c$ is

$$\mathrm{Supp}(c, g) \;=\; \mathbb{P}\big(g(x) = 1 \mid y(x) = c\big), \tag{24}$$

i.e., the fraction of class-$c$ examples on which the rule fires. To make precision insensitive to class imbalance, we report a balanced one-vs-rest quantity:

$$\mathrm{Prec}_{\mathrm{bal}}(c, g) \;=\; \frac{\mathbb{P}\big(g(x) = 1 \mid y(x) = c\big)}{\mathbb{P}\big(g(x) = 1 \mid y(x) = c\big) + \mathbb{P}\big(g(x) = 1 \mid y(x) \neq c\big)}. \tag{25}$$

Intuitively, $\mathrm{Supp}(c, g)$ measures how frequently $g$ is activated within class $c$, while $\mathrm{Prec}_{\mathrm{bal}}(c, g)$ measures how selectively $g$ concentrates on class $c$ versus all other classes. In Table 2, "Support" and "Precision" correspond to (24) and (25), estimated by empirical frequencies on the evaluation split.

**Proto-CAM vs. Rule-CAM (Figure 5).** Both visualizations in Figure 5 are obtained with standard Grad-CAM; the only difference lies in the scalar score that is explained. For a client $i$, input $x$, and class $c$, let $\Phi_i(x)$ denote the backbone feature map at the target layer and $\mathrm{GAP}(\Phi_i(x))$ its global-average pooled vector. Let $p_c$ be the global class prototype in the shared prototype space. The prototype-based score is defined as

$$s_{\mathrm{proto}}(x) \;=\; \cos\big(\mathrm{GAP}(\Phi_i(x)),\, p_c\big), \tag{26}$$

and the Proto-CAM heatmap is obtained by applying Grad-CAM to $s_{\mathrm{proto}}(x)$ at the same target layer.

For Rule-CAM, we feed the slot representation of $x$ into the rule network and collect the logic activations $\{r_k(x)\}$ corresponding to the positive rules associated with $(c, a)$. We then form the rule-based score

$$s_{\mathrm{rule}}(x) \;=\; \frac{1}{|H_{c,a}^+|} \sum_{k \in H_{c,a}^+} r_k(x), \tag{27}$$

and apply the same Grad-CAM procedure to $s_{\mathrm{rule}}(x)$.

### D.5. Additional Results

**Convergence Speed.** Figure 9 shows that RFL converges faster and more smoothly than prototype-based baselines, while reaching a higher converged accuracy. By using rules as the knowledge-sharing carrier, RFL provides a more architecture-friendly discriminative constraint, which offers clear guidance in the early training stage and mitigates the semantic mismatch and oscillation caused by naively averaging heterogeneous prototypes.

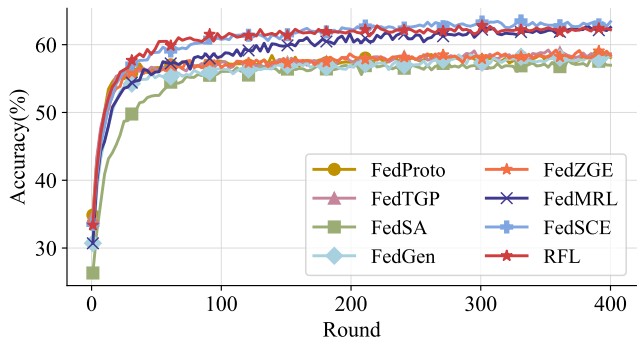

*(a)* Average accuracy on CIFAR-100(*Pat*).

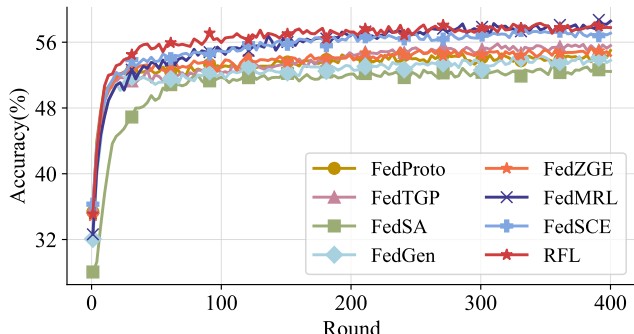

*(b)* Average accuracy on CIFAR-100($Dir_{0.1}$).

*Figure 9.* Convergence curves of all methods on CIFAR-100. The accuracy at each round is the average test accuracy over all clients.

*Table 4.* Effect of boundary augmentation on CIFAR-100. Gain is computed relative to the setting without boundary augmentation.

| Method | Accuracy (%) ↑ | Gain |
|---|---|---|
| RFL w/o boundary augmentation | 56.81 | – |
| RFL | 58.25 | +1.44 |

Compared with proxy-model-based methods, RFL also converges faster in the early stage but tends to saturate later. This is because the rule knowledge is distilled from class-level summaries, which are compact and transferable but have limited capacity. In contrast, methods such as FedSCE and FedMRL train proxy models with richer local supervision and may obtain more comprehensive teacher signals, potentially achieving higher final accuracy in some settings. However, this advantage comes with substantially higher local computation and communication overhead. Overall, the rule network serves as a stable collaboration carrier under heterogeneity without disrupting effective local training, offering a more controllable trade-off between efficiency and performance.

**Effect of Boundary Augmentation.** Boundary augmentation is introduced to prevent the rule network from simply memorizing class-center prototypes and to encourage discriminative rule learning near inter-class boundaries. We ablate this component on CIFAR-100. As shown in Table 4, removing boundary samples decreases the final average accuracy from 58.25% to 56.81%, yielding a 1.44-point drop. This confirms that the gain of RFL does not merely come from increasing the number of server-side samples, but from providing boundary-aware training signals that help the rule network learn more discriminative class rules.

**Architecture-wise Gains.** Figure 10 groups the overall results by backbone and reports each method's average accuracy gain over local-only training. RFL achieves the largest and most balanced improvements across both tasks and all model groups. In contrast, prototype-based methods exhibit a pronounced gain collapse on ViT clients. Under strong heterogeneity, different backbones induce substantially different feature geometries and decision criteria, so collaboration through shared prototypes often yields unbalanced benefits and may even cause negative transfer for minority architectures. By using

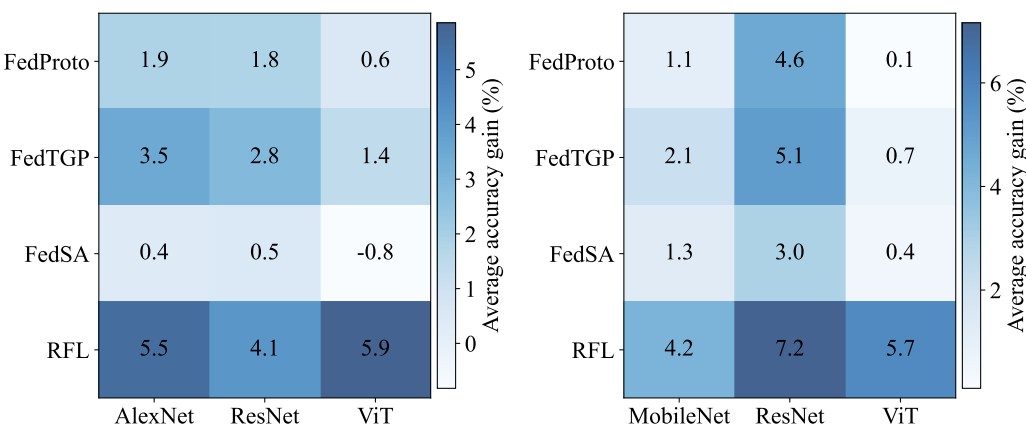

*(a)* Average accuracy gain on CIFAR-100($Dir_{0.1}$).   *(b)* Average accuracy gain on Tiny-ImageNet($Dir_{0.1}$).

*Figure 10.* Architecture-wise average accuracy gain (%) over local-only training. Each cell reports the improvement in average test accuracy achieved by a method compared to training each client independently (local-only), averaged over all clients within the same backbone architecture. Left: CIFAR-100 (AlexNet/ResNet/ViT). Right: Tiny-ImageNet (MobileNet/ResNet/ViT). Higher values indicate larger gains from cross-client collaboration, while negative values indicate degradation.

*Table 5.* Classification accuracies (%) under different data partitions. To reduce the degree of model heterogeneity, we use only convolutional architectures (AlexNet, ResNet-10, and ResNet-18).

| Method Group | Datasets | | *CIFAR-100* | |
|---|---|---|---|---|
| | Settings | *Pat* | $Dir_{0.1}$ | $Dir_{0.3}$ |
| Proxy-model methods | FedGen | 64.08 | 59.65 | 42.64 |
| | FedZGE | 64.51 | 59.54 | 42.63 |
| | FedMRL | 62.03 | 58.06 | 41.35 |
| | FedSCE | 66.73 | 62.71 | 46.77 |
| Prototype methods | FedProto | 63.13 | 59.80 | 42.58 |
| | FedTGP | 64.88 | 61.61 | 45.25 |
| | FedSA | 64.36 | 60.45 | 43.31 |
| Ours | RFL | 67.51 | 63.56 | 47.45 |

cross-architecture transferable rules as the collaboration carrier, RFL is more likely to deliver consistent positive gains across backbone groups. As a result, RFL provides more reliable cross-client collaboration under model heterogeneity, enabling clients with diverse architectures to cooperate stably and improve together.

**Additional Heterogeneity Studies.** Table 5 further reduces model heterogeneity on CIFAR-100 by using only convolutional backbones, i.e., AlexNet, ResNet-10, and ResNet-18, and compares different methods under three data partitions. RFL continues to exhibit overall advantages, indicating that its effectiveness is not limited to extreme heterogeneity, such as settings including ViT. These results further show that RFL also provides consistent gains under milder model heterogeneity, demonstrating the robustness of the framework.

