# OpenReview forum: "Towards Rule-Based Knowledge Sharing in Federated Learning"
_ICML.cc/2026/Conference — ICML 2026 regular_

### Official Review · Reviewer_reDi · 2026-02-27

**Soundness:** 3
**Presentation:** 3
**Significance:** 3
**Originality:** 3
**Overall Recommendation:** 4
**Confidence:** 3

**Summary:**

This paper studies an important problem that data heterogeneity and model heterogeneity simultaneously exist, and proposes RFL that uses interpretable class decision rules as the knowledge-sharing carrier. RFL introduces selective rule absorption, broadcasting only high-convergence rules. Experiments show that RFL achieves superior accuracy–communication–computation trade-off.

**Compliance With Llm Reviewing Policy:**

Affirmed.

**Final Justification:**

The authors' rebuttal has addressed my main concerns. Using trainable rule network as the knowledge carrier is interesting. I would like to keep my current score.

**Key Questions For Authors:**

How does "AND/OR composition" of boolean atoms constitute the decision grounds for each class?

**Limitations:**

Uploading class-level feature prototypes may leak privacy.

**Strengths And Weaknesses:**

**Strengths**

1. Using trainable rule network as the knowledge carrier is interesting, enabling cross-device rule learning with different architectures.

2. Encoding knowledge as class-wise rules enhances the interpretability while avoiding the averaging of semantically mismatched features.

3. A diverse set of backbone architectures are adopted across different clients and datasets, demonstrating the robustness under the heterogeneity of model architecture.

4. Compared with transmitting the whole model parameters, the server only broadcasts the parameters of rule network and rule indices, so the communication cost is reduced.



**Weaknesses**

1. The motivation of logic layer is missing. It is unclear how the AND/OR combinations of boolean atoms constitute the decision grounds for each class.

2. The performance of RFL is somewhat sensitive to the hyperparameter setting.

3. The ablation of boundary augmentation is missing. How does the augmented set $D_{bnd}$ contribute to the boundary discrimination?

4. It is unclear what $F_i(\theta_i)$ means in Eq.(2).

---

> ### Author Rebuttal · Authors · 2026-03-31
>
> We sincerely thank the reviewer for the careful reading and comments.
>
> >Q1. The motivation of logic layer is missing. It is unclear how the AND/OR combinations of boolean atoms constitute the decision grounds for each class.
>
> R1. The motivation for the logic layer is twofold. First, the atoms produced by SoftBin only indicate whether a coordinate crosses a threshold. They are single-dimensional, local, and weak evidence, and are therefore insufficient for class discrimination on their own. Second, what is truly stable and shareable across heterogeneous clients is usually not an individual value, but which pieces of evidence should co-occur, and which different evidence patterns can all support the same class.
>
> We therefore introduce the logic layer to organize scattered atoms into compositional rules. AND units learn co-occurrence patterns that must be satisfied jointly, while OR units learn alternative support patterns that can substitute for one another. For example, let $A, B, C, D$ be four atoms, each representing a simple threshold condition. For class $c$, the rule $((A \land B) \lor (C \land D))$ means that either the co-occurrence of $A$ and $B$, or that of $C$ and $D$, provides positive support for class $c$. The shared classifier then selects the subset that consistently contributes positively to that class. These high-weight positive rules form its decision grounds. Thus, the logic layer compresses weak atomic evidence into class-level discriminative patterns that can be selected and broadcast.
>
> >Q2. The performance of RFL is sensitive to hyperparameter setting.
>
> R2. Hyperparameters do affect RFL, but the method remains stable over a moderate range and degrades only at extreme values, as is common for methods with regularization and distillation. Figure 7 in the paper shows that, for the distillation weight $\alpha_{kd}$, rule weight $\alpha_{ru}$, and margin parameter $m$, performance mainly drops only when the value is too small or too large, while remaining stable over a relatively wide middle range. This is a mild fluctuation rather than a sharp collapse.
>
> >Q3. The ablation of boundary augmentation is missing. How does the augmented set  contribute to the boundary discrimination?
>
> R3. Boundary augmentation is not meant merely to increase server-side samples, but to prevent the rule network from memorizing class-center patterns and to encourage learning discriminative boundary structures between neighboring classes. This motivation is already stated in the main text. We therefore additionally construct $D_{bnd}$ to force the rule network to learn discriminative rules that separate inter-class boundaries. Table R5 shows that this ablation is consistent with the above explanation. Without boundary augmentation, the final accuracy is 56.81, while full boundary augmentation improves it to 58.25. This suggests that the gain does not come merely from having more augmented samples, but from the fact that boundary-aware training signals help the rule network learn more discriminative rules.
>
> Table R5. Effect of boundary augmentation on CIFAR-100. Gain vs. No-boundary shows relative accuracy gain.
>
> \\begin{array}{lcc} \\hline \\text{Setting} & \\text{Avg. Acc. (\\%)} \\uparrow & \\text{Gain vs. No-boundary} \\\\ \\hline \\text{No boundary (prototypes only)} & 56.81 & 0.00 \\\\ \\text{Full boundary (ours)} & 58.25 & +1.44 \\\\ \\hline \\end{array}
>
> >Q4. It is unclear what means in Eq.(2).
>
> R4. Eq. (2) is intended as a high-level template for heterogeneous FL, rather than the executable training objective of RFL. Specifically, $K$ denotes a shareable knowledge carrier, $\Phi$ denotes the mechanism by which a client absorbs collaborative knowledge from $K$, $F_i(\theta_i)$ denotes the client’s self-learning objective, and $J(\cdot)$ is an abstract global generalization objective over these components. In our method, this template is instantiated as $K = R$, where $R$ is the server-side rule network, and $\Phi$ is realized by rule regularization and rule distillation. The concrete local objective is therefore Eq. (12), i.e., $L_i^{ce} + \alpha_{ru} L_i^{rule} + \alpha_{kd} L_i^{kd}$.
>
> >Q5. Uploading class-level feature prototypes may leak privacy.
>
> R5. RFL uploads class-level prototype statistics rather than raw samples or per-sample features, so the underlying sample data itself remains undisclosed. More accurately, RFL has the same privacy boundary as existing prototype-based FL methods. The exchanged information is coarser-grained and therefore more privacy-friendly than directly sharing raw data or per-sample features, but it does not eliminate the residual risk of prototype sharing. We leave the integration of stricter privacy-preserving mechanisms to future work.

---

> > ### Author Rebuttal · Reviewer_reDi · 2026-04-02
> >
> > Thanks for authors' rebuttal. I have no more concerns. I will keep my score.

---

> > > ### Author Response · Authors · 2026-04-03
> > >
> > > Dear Reviewer reDi,
> > >
> > > We sincerely thank you for reading our rebuttal. We are very grateful that our additional clarifications and experimental results were able to eliminate your concerns.
> > >
> > > We also thank you for your constructive comments and your positive evaluation of the paper. These comments have helped us further improve the paper, and we will carefully adopt them.
> > >
> > > Thank you again for your valuable comments and support.
> > >
> > > Authors

---

### Official Review · Reviewer_bPLE · 2026-03-08

**Soundness:** 3
**Presentation:** 3
**Significance:** 3
**Originality:** 3
**Overall Recommendation:** 4
**Confidence:** 3

**Summary:**

This paper addresses the challenge of knowledge sharing in heterogeneous federated learning, where existing solutions directly align or average features, inducing semantic confusion and distorted decision boundaries. The authors propose rule-based federated learning (RFL) that enables heterogeneous collaboration at the rule level. RFL first encodes client prototypes into interpretable rules to avoid incompatible representations and then leverages a sparse rule set and selective rules absorption to simultaneously reduce communication costs and mitigate negative transfer. Experiments on CIFAR-10/100 and Tiny-ImageNet with diverse architectures show improved accuracy-communication trade-offs versus prototype sharing and substantially lower communication than proxy-model methods.

**Compliance With Llm Reviewing Policy:**

Affirmed.

**Final Justification:**

I thank the authors for their detailed and thoughtful responses, which have clarified several of my concerns. After carefully considering the rebuttal alongside the original submission, I will maintain my original score.

**Key Questions For Authors:**

The paper is well-written, but there are some minor typographical issues. For example, in the Abstract, "often face" should be "often faces," and in Section 3.2.2, the subject is missing in "employ logical layers."
Across all experiments, the three heterogeneous architectures are represented by clients in a fixed 3:4:3 ratio. If this distribution becomes highly imbalanced, e.g., 8:1:1, could the rule network become biased toward the dominant architecture’s feature distribution, potentially leading to degraded performance for the minority clients?

**Limitations:**

Please discuss the limitations and potential negative societal impact of this paper in a separate session.

**Strengths And Weaknesses:**

Claims and Evidence:
The theoretical claims are well-supported; the proofs in the appendix appear correct and complete.

Experimental Designs or Analyses:
The experimental design is generally sound. However, the number of clients in the paper is fixed and relatively small, which may be insufficient for large-scale federated scenarios, e.g., 1000+ clients and 100+ categories. It is suggested to consider how the proposed approach scales in such settings, particularly whether the rule set could grow substantially and thus affect the claimed communication efficiency.
In addition, all feature extractors in the experiments are constrained to output representations with the same dimensionality. It would be helpful for the authors to discuss the applicability of the method when this condition is hard to satisfy, e.g., when architectures naturally produce features of very different dimensions and require additional projection, as well as the potential impact on performance.

Relation to Broader Scientific Literature:
This paper aligns well with the literature on Heterogeneous Federated Learning (HtFL) and provides a detailed exploration of the advantages and limitations of existing mainstream knowledge carriers, such as proxy models, nested sub-models, and shared prototypes. The authors make an innovative contribution by introducing rule learning and symbolic reasoning, e.g., neural logic machines from within a single model, into cross-client collaboration in federated learning.

Other Strengths and Weaknesses:
This paper introduces interpretable logical rules as a knowledge-sharing carrier for heterogeneous federated learning. Through a two-phase design involving rule knowledge construction on the server side and selective rule absorption on the client side, the approach balances classification accuracy with communication and computation efficiency. Additionally, the use of AND/OR logical layers to combine boolean rules enhances the interpretability of the cross-architecture decision-making process.
However, it is suggested to include a discussion of privacy, which is one of the core motivations for federated learning. In particular, since the class prototypes uploaded by clients are averages of raw features, it would be helpful to clarify whether these aggregated statistics could still reveal sensitive information about the underlying training data.

---

> ### Author Rebuttal · Authors · 2026-03-31
>
> We thank the reviewer for the positive evaluation and constructive suggestions.
>
> >Q1. Large-scale scalability is not covered: under 1000+ clients / 100+ categories, could the communication efficiency be affected?
>
> R1. The 100+ category regime is partly reflected in our experiments via Tiny-ImageNet (200 classes). As the number of clients $N$ grows, total communication of almost all server-mediated FL methods increases; the key difference is the dominant term. For RFL, the uplink remains prototype-level, while the downlink scales as $O(N(|\theta_{\mathrm{rule}}|+\sum_c k_c))$, mainly transmitting a lightweight rule network and sparse retained-rule indices. By contrast, transmitting a full proxy model incurs a much larger cost $O(N|\theta_{\mathrm{proxy}}|)$. Moreover, RFL keeps only high-coverage positive rules, and each client activates only those relevant to its local classes. Thus, although transmitted rules grow with scale, they do not grow with the full rule library.
>
> From Table 1, extrapolating the same protocol from 30 to 1000 clients gives 1.40/1.54 GB per round on CIFAR-100/Tiny-ImageNet for RFL, versus 8.85/9.04 GB for FedMRL, still 84.2%/82.9% lower. A full 1000+ client experiment is resource-intensive and difficult to complete rigorously within the rebuttal window; we will prioritize it in future validation.
>
> >Q2. When backbone output dimensions differ substantially and additional projection is required, is the method still applicable, and what is the potential impact on performance?
>
> R2. In our experiments, we directly unify each backbone’s final representation dimension, without an extra projection head. This same backbone-output setting is used across the compared methods, ensuring a fair comparison. When native output dimensions differ more, an additional projection may be needed. RFL still applies, because server-side rule learning only requires unified-dimensional class prototypes as input.
>
> Constructing prototypes from projected features does not break the RFL pipeline. Performance then depends on whether the projection preserves discriminative information. If it mainly builds a shared interface and reduces architecture-specific noise, it may even help RFL by making class summaries more comparable. If compression is too strong and creates an information bottleneck, it may weaken prototype quality and hurt performance.
>
> >Q3. Could prototypes still leak sensitive information about the training data?
>
> R3. RFL uploads class-level prototype statistics rather than raw samples or full model updates, so the server receives class-level statistics rather than underlying sample data. Prototypes are among the most common collaboration carriers in heterogeneous FL. More precisely, RFL has the same privacy boundary as existing prototype-based FL methods. The exchanged information is coarser-grained and thus more privacy-friendly than raw data or per-sample representations, but it does not eliminate the residual risk of prototype sharing. As an additional check, we perturb uploaded prototypes on CIFAR-100 before transmission. When the noise magnitude is 0.02 and 0.05, the final average accuracy decreases from 58.25 to 57.84 and 57.12, with drops of only 0.4 and 1.1. This suggests that RFL remains compatible with privacy-preserving mechanisms, and stronger privacy protection is a natural future direction.
>
> >Q4. Under imbalanced architecture proportions, could the dominant architecture dominate rule learning and degrade minority clients?
>
> R4. RFL already includes mechanisms to address this risk. For each class, the server groups prototypes by architecture label and then performs balanced sub-sampling, to avoid any single architecture dominating rule learning. In addition, SoftBin uses architecture-conditioned affine calibration, and the logic layers retain architecture codes to absorb remaining differences. In other words, RFL does not simply pool all prototypes to train rules, but explicitly accounts for majority-architecture dominance.
>
> As shown in Table R4, when the ratio changes from 3:4:3 to 8:1:1, RFL drops by only 0.93, and under 8:1:1 the average accuracy of minority clients remains clearly higher than FedProto. Thus, performance decreases under extreme imbalance, but minority clients in RFL do not show clear suppression by the majority architecture.
>
> Table R4. Robustness under imbalanced architecture on CIFAR-100. Minority Avg. Acc. denotes the average accuracy of minority architectures.
>
> \\begin{array}{lccc} \\hline \\text{Method} & \\text{3:4:3 Acc.} & \\text{8:1:1 Acc.} & \\text{8:1:1 Minority Avg. Acc.} \\\\ \\hline \\text{FedProto} & 54.52 & 52.46 & 50.31 \\\\ \\text{RFL} & 58.25 & 57.32 & 56.69 \\\\ \\hline \\end{array}
>
> >Q5. There are also some minor wording issues.
>
> R5. Thank you for pointing out these issues. We will correct errors such as “often face” in the abstract, add the missing subject in Section 3.2.2, and check for similar issues elsewhere.

---

> > ### Author Rebuttal · Reviewer_bPLE · 2026-04-02
> >
> > Thank you for the authors' response, which has addressed most of my questions. I will maintain or raise my score.

---

> > > ### Author Response · Authors · 2026-04-03
> > >
> > > Dear Reviewer bPLE,
> > >
> > > We sincerely thank you for taking the time to conduct a careful review of our paper and for giving a positive evaluation. We are deeply encouraged to know that our additional clarifications and experiments have helped address your concerns.
> > >
> > > We also thank you for your constructive suggestions. Your feedback is of great value in improving the quality of the paper, and we will carefully incorporate it into the revised manuscript.
> > >
> > > Thank you again for your support.
> > >
> > > Authors

---

### Official Review · Reviewer_L5eh · 2026-03-11

**Soundness:** 3
**Presentation:** 3
**Significance:** 3
**Originality:** 3
**Overall Recommendation:** 4
**Confidence:** 4

**Summary:**

This paper focuses on the heterogeneous federated learning scenario and proposes a rule based federated learning framework called RFL. In strongly heterogeneous settings, directly sharing prototypes can easily cause semantic confusion, while proxy model based methods usually bring higher cost. This paper uses interpretable class rules as the communication carrier. The server trains a rule network based on the class prototypes and architecture labels uploaded by clients, and then broadcasts the filtered rules and the rule network to the clients. The experimental results prove that the method has strong advantages.

**Compliance With Llm Reviewing Policy:**

Affirmed.

**Final Justification:**

The author has solved my problem and I support accepting it.

**Key Questions For Authors:**

Q1. The server side rule network is mainly built on class level summaries such as class prototypes, and the appendix also points out that although this kind of rule knowledge is compact and transferable, its accuracy may be lower than that of training an additional proxy model because the capacity of class prototypes is far more limited than that of a proxy model. Could the authors explain more clearly in which scenarios this capacity limit is most evident, and whether richer intra class statistics or stronger rule carriers may be needed in the future for improvement?

Q2. The paper mentions that for rule training, all feature extractors must output representations of the same dimension. Could the authors clarify whether this is achieved through an additional projection head or by directly modifying the final layer of the backbone? Furthermore, was this same design applied to all baselines to ensure a fair comparison?

**Limitations:**

yes

**Strengths And Weaknesses:**

Strengths

1. This paper redefines the collaboration mechanism in heterogeneous federated learning from the perspective of the knowledge carrier, replacing traditional prototypes or proxy models with rules. The problem is well motivated, and the overall novelty is strong.

2. Under fully heterogeneous architectures, the method can still stably improve accuracy. At the same time, it does not require transmitting proxy models, resulting in lower communication cost, and it also has good rule interpretability.

3. The paper provides theoretical analysis from the perspectives of rule space generalization and selective absorption mechanisms, giving the method a certain degree of theoretical support.

Weaknesses

1. The paper mentions that, for rule training, all feature extractors finally output feature representations of the same dimension. This setting is reasonable in the experiments, but the authors are advised to explain more clearly in the main text whether this is achieved through a projection head or by directly unifying the last layer of the backbone, and whether this design is also used for the baselines. This would help avoid readers misunderstanding that cross architecture rule sharing relies on additional adaptation that has not been fully explained.
2. The interpretability of the paper is reflected in the fact that its rules are readable. In essence, they are threshold combinations on latent slot dimensions. This is certainly more structured than an invisible vector, but it is still somewhat far from human understandable semantic rules.
3. The method still takes class prototypes as the core input, and class means are insufficient to fully characterize intra class multimodality and complex boundaries. Therefore, the expressive capacity of the rule carrier has a natural upper bound. The appendix also discusses that the method is more likely to saturate in the later stage, and its final accuracy may be lower than that of some proxy model methods. This limitation should be stated in the main text.

---

> ### Author Rebuttal · Authors · 2026-03-31
>
> We sincerely thank the reviewer for the positive evaluation of our paper and for the constructive comments.
>
> >Q1. How is the unified feature-output dimension implemented, and was the same design applied to all baselines to ensure a fair comparison?
>
> R1. In our experiments, the unified feature dimension is achieved by directly setting the output dimension of the last layer of each backbone, without introducing any additional projection head. The purpose is neither to weaken model heterogeneity nor to add hidden adaptation specifically for RFL, but to provide a consistent shared interface for all methods that interact through class-level representations. In fact, this is also a common setting in prototype-based heterogeneous federated learning. Accordingly, in our experiments, all baselines that rely on class prototypes for collaboration adopt the same interface. For proxy-model methods such as FedMRL and FedSCE, which do not rely on a prototype interface, we keep the same heterogeneous backbones and training budget, without additionally imposing such an interface, so as to ensure a fair comparison.
>
> >Q2. The current rules are more like structured, readable threshold combinations, and still remain somewhat far from human-understandable semantic rules.
>
> R2. More precisely, the rules in our paper should be understood as structured and inspectable decision rules, rather than fully human-semantic natural-language rules. At the same time, we would like to emphasize that RFL is still more interpretable than simple prototypes and black-box auxiliary models.
>
> Its interpretability mainly comes from two aspects. First, prototypes are discretized by SoftBin into Boolean atoms, then composed by AND/OR logic layers into inspectable rule units, and finally filtered into class-specific positive rules. Therefore, the decision structure is explicit and traceable. Second, what each client absorbs locally are also class-selected positive rules, rather than black-box prototype vectors or proxy-model outputs. In future work, we will further explore combining such rules with natural language models, so as to gradually move toward more human-understandable natural-language rules.
>
> >Q3. Since the method still takes class prototypes as the core input, the expressive capacity of the rule carrier has a natural upper bound, and this limitation should be acknowledged.
>
> R3. RFL places greater emphasis on achieving competitive accuracy under heterogeneity with lightweight overhead. Its goal is therefore to attain a better accuracy-efficiency trade-off, rather than unconditionally pursuing the highest final accuracy. Proxy-model methods can achieve higher final accuracy in some settings because their auxiliary models directly absorb richer client-side sample-level information and thus possess stronger representational capacity. However, this does not mean that proxy-model methods are uniformly stronger. In our main results, only FedMRL is slightly above RFL on CIFAR-100 Dir(0.1), whereas on the more complex Tiny-ImageNet task RFL achieves the best accuracy among all compared methods, suggesting that the current rule carrier already provides a favorable balance between expressiveness and efficiency. At the same time, this accuracy advantage, when it appears, usually comes with substantially higher computation and communication overhead, and in our experiments, such extra cost does not yield commensurate performance gains.
>
> Although moderately increasing the capacity of the rule carrier can improve accuracy to some extent, it also weakens the lightweight advantage of RFL. A natural direction for further strengthening the method is therefore to incorporate richer intra-class statistics, so that the server can receive more informative client knowledge, or to design a stronger server-side rule carrier, thereby further enhancing the expressive capacity of the rule carrier.

---

> > ### Author Rebuttal · Reviewer_L5eh · 2026-04-01
> >
> > Thank you for the authors' response, which has addressed my questions. I champion this paper and will maintain or raise my score.

---

> > > ### Author Response · Authors · 2026-04-03
> > >
> > > Dear Reviewer L5eh,
> > >
> > > We sincerely thank you for your valuable comments and for the time and care you devoted to evaluating our work. We are deeply encouraged to know that our additional response has helped eliminate your concerns.
> > >
> > > We also greatly appreciate your constructive suggestions, which are of great value in improving and strengthening the paper, and we will carefully incorporate them.
> > >
> > > Thank you again for your valuable feedback.
> > >
> > > Authors

---

### Official Review · Reviewer_e5HP · 2026-03-12

**Soundness:** 3
**Presentation:** 3
**Significance:** 2
**Originality:** 2
**Overall Recommendation:** 5
**Confidence:** 3

**Summary:**

This paper addresses heterogeneous federated learning with data and model heterogeneities and proposes a rule-sharing-based mechanism, RFL. Specifically, each client holds heterogeneous models and uploads class prototypes as feature evidence, and the server trains an interpretable rule network to differentiate class features. Extensive experiments demonstrate the effectiveness of the proposed method.

**Compliance With Llm Reviewing Policy:**

Affirmed.

**Final Justification:**

The paper is technically solid. I appreciate the authors' efforts in providing additional experimental results to support the claim that RFL  consistently outperforms the most powerful proxy-model method at a slightly higher communication cost. In my opinion, this work would be valuable to the ICML community and to researchers in federated learning. Therefore, I vote for acceptance.

**Key Questions For Authors:**

See **Weaknesses** specific to the first and third points.

**Limitations:**

See **Weaknesses** specific to the second point.

**Strengths And Weaknesses:**

**Strengths:**
1. This paper explores an important problem in FL where both models and data are heterogeneous among the clients.
2. The idea is interesting because the server learns and broadcasts a rule network instead of relying on conventional parameter aggregation or a proxy deep model.
3. The paper provides a fairly rigorous theoretical analysis.
4. The empirical study is fairly extensive, including several baselines and accuracy-efficiency comparisons.

**Weaknesses:**
1. The related work section could be strengthened. The paper already reviews heterogeneous FL through prototype-based, proxy-model-based, and other representation-sharing methods, but it does not sufficiently discuss the connection to split learning. In this method, clients extract class-level feature prototypes and send them to the server, which then trains the global rule carrier from these uploaded representations. Although this is not identical to standard split learning, it is still close enough that the paper should discuss the relation more carefully. In addition, the paper should more explicitly compare with closely related prototype-based methods such as [1] and [2], and clarify the distinction from existing class-prototype or global-prototype learning approaches.
2. I also have concerns about the privacy claim. The paper states that clients upload class-level feature prototypes together with architecture labels, and the server constructs supervised samples from them to train the rule network. While this is certainly more privacy-friendly than sharing raw samples, I am not convinced this fully resolves privacy risk. In particular, since the server receives class-specific representations, it seems possible that these prototypes could still leak information about client data or enable reconstruction or attribute inference, especially when local class distributions are sparse. The appendix frames the privacy argument mainly as "structural privacy by design," rather than providing a formal leakage analysis, so I think this limitation should be acknowledged more explicitly.
3. From the experimental results, RFL does not always dominate the strongest proxy-model methods in pure final accuracy. The paper itself notes that methods such as FedSCE and FedMRL can exploit richer local supervision through auxiliary models and may achieve higher final accuracy, while RFL offers a better communication-efficiency trade-off. This is a reasonable design choice, but it also raises an important question: could RFL achieve better accuracy if the method were allowed a slightly larger communication budget or a more expressive server-side rule carrier? I think the paper would be stronger if it discussed this trade-off more directly, rather than emphasizing efficiency alone.

**References:**
[1] Taming Cross-Domain Representation Variance in Federated Prototype Learning with Heterogeneous Data Domains
[2] FedTGP: Trainable Global Prototypes with Adaptive-Margin-Enhanced Contrastive Learning for Data and Model Heterogeneity in Federated Learning

---

> ### Author Rebuttal · Authors · 2026-03-31
>
> We thank the reviewer for the positive assessment and valuable comments.
>
> >Q1. The related work section could be strengthened.
>
> R1. RFL is similar to split learning only in that the server learns from client-uploaded summaries; the training paradigms differ. Standard split learning partitions a task network at a cut layer and continues training the same network on the server using sample-level activations. In contrast, RFL neither splits the task network nor performs cross-device joint backpropagation; clients upload only class-level prototypes, and the server trains an independent rule carrier.
>
> The key distinction from [1] and [2] is what shared knowledge is ultimately learned and returned. Method [1] enriches prototypes via clustered prototypes and an $\alpha$-sparsity loss, while [2] learns trainable global prototypes; however, both still collaborate in prototype space. By contrast, RFL uses prototypes only as input evidence and transforms them into a rule carrier shareable across architectures. We additionally compare with [1], and Table R1 shows it still underperforms RFL. This suggests that under strong heterogeneity, the bottleneck is not insufficient prototype richness, but cross-architecture semantic incompatibility. Transforming prototype evidence into a more transferable rule space is therefore more effective in alleviating this issue. We will clarify this relation in the revision.
>
> Table R1. Results on CIFAR-100.
>
> \\begin{array}{lcc} \\hline \\text{Method} & \\text{Acc. }\\text{Dir}(0.1) & \\text{Acc. }\\text{Dir}(0.3) \\\\ \\hline \\text{FedPLVM [1]} & 54.12 & 38.65 \\\\ \\text{RFL} & 58.25 & 43.80 \\\\ \\hline \\end{array}
>
> References:
>
> [1] Wang L, et al. Taming Cross-Domain Representation Variance in Federated Prototype Learning with Heterogeneous Data Domains, NeurIPS, 2024.
>
> [2] Zhang J, et al. FedTGP: Trainable Global Prototypes with Adaptive-Margin-Enhanced Contrastive Learning for Data and Model Heterogeneity in Federated Learning, AAAI, 2024.
>
> >Q2. I also have concerns about the privacy claim.
>
> R2. In heterogeneous FL, prototypes are among the most common cross-architecture knowledge carriers. RFL uploads class-level prototypes, which is consistent with existing prototype-based heterogeneous FL methods in that they mainly exchange class-level distributional information. Therefore, its privacy exposure is clearly smaller than raw data or per-sample representations. This does not mean there is no residual privacy risk, but such risk is better characterized at the class level, since low-dimensional prototypes averaged over multiple samples are inherently harder to invert.
>
> As a supplement, we test a simple privacy protection mechanism by adding noise before prototype upload. As shown in Table R2, RFL remains robust under additional privacy protection, with little accuracy drop. Therefore, a stricter formal leakage analysis, and a more systematic integration with privacy-preserving mechanisms, will be added as future work.
>
> Table R2. Effect of prototype sanitization on CIFAR-100.
>
> \\begin{array}{lccc} \\hline \\text{Setting} & \\text{Uploaded prototype processing} & \\text{Avg Acc. (\\%)}\\uparrow & \\text{Drop vs. Raw} \\\\ \\hline \\text{Raw} & \\text{no noise} & 58.25 & 0.0 \\\\ \\text{small noise} & \\text{noise }=0.02 & 57.84 & -0.4 \\\\ \\text{stronger noise} & \\text{noise }=0.05 & 57.12 & -1.1 \\\\ \\hline \\end{array}
>
> >Q3. From the experimental results, RFL does not always dominate the strongest proxy-model methods in pure final accuracy.
>
> R3. RFL is designed to balance accuracy and cost. On Tiny-ImageNet, it achieves the best accuracy, outperforming the strongest proxy-model baseline by up to 0.84 points while using much lower communication. This shows that proxy-model advantages are not unconditional, but depend on whether the proxy model has sufficient capacity; in contrast, the rule carrier remains robust under stronger heterogeneity and higher task complexity. Table R3 further shows that RFL can improve further, and that enlarging the carrier is more effective than increasing rule broadcast, suggesting that its main limitation is carrier expressiveness rather than bandwidth alone. Therefore, stronger rule carriers or richer intra-class summaries are promising directions for improving RFL. Meanwhile, for proxy-model methods to further improve on harder tasks, they often also need a larger proxy model, increasing both communication and local training cost.
>
> Table R3. Cost scaling study on CIFAR-100.
>
> \\begin{array}{lcccc}
> \\hline
> \\text{Method} & \\text{Rule carrier size (AND/OR)} & \\text{Broadcast budget} & \\text{Avg Acc. (\\%) }\\uparrow & \\text{Comm. (MB/round) }\\downarrow \\\\
> \\hline
> \\text{FedMRL} & - & - & 58.43 & 265.36 \\\\
> \\text{RFL} & 150 / 150 & 1.0\\times & 58.25 & 41.98 \\\\
> \\text{RFL-large-carrier} & 250 / 250 & 1.0\\times & 58.87 & 72.50 \\\\
> \\text{RFL-large-budget} & 150 / 150 & 2.0\\times & 58.30 & 46.21 \\\\
> \\hline
> \\end{array}

---

> > ### Author Rebuttal · Reviewer_e5HP · 2026-04-01
> >
> > Thank you for your response. Most of my concerns are addressed. I will keep my positive score unchanged.

---

> > > ### Author Response · Authors · 2026-04-03
> > >
> > > Dear Reviewer e5HP,
> > >
> > > Thank you very much for taking the time to read our rebuttal and evaluate our paper. We sincerely appreciate that our additional clarifications and experimental results have effectively addressed your concerns.
> > >
> > > We also sincerely thank you for your constructive comments. Your feedback has been very helpful in improving the paper, and we will carefully incorporate these suggestions into the revised manuscript.
> > >
> > > Thank you again for your valuable time and support.
> > >
> > > Authors

---

### Decision · Program_Chairs · 2026-04-30

**Decision:**

Accept (regular)

**Comment:**

The paper received mixed but overall positive reviews, with three weak accepts and one weak reject. This paper proposes MetaMoE, a framework for privacy-constrained unification of independently trained domain experts into a single deployable Mixture-of-Experts model, using relevance-weighted proxy selection from public data, proxy-aligned expert training, and a context-aware router. Reviewers appreciated the practical importance of the problem, the clear and well-structured pipeline, and the broad empirical evaluation across both CV and NLP tasks. The method was viewed as technically reasonable and generally effective, with thorough ablations supporting the contributions of proxy selection, alignment, and routing design.
The main concerns centered on the scope of the privacy claim and the lack of a stronger formal privacy guarantee. In particular, one reviewer remained unconvinced that sharing routing vectors derived from private-data embeddings fully rules out domain-level or distributional leakage, even after rebuttal. Additional concerns included possible dependence on public data quality and domain overlap, limited novelty of some components such as the context-aware router, and evaluation at relatively modest scale. That said, the rebuttal substantially strengthened the paper through added experiments on proxy mismatch, alternative public datasets, larger client settings, and seed variance, and most reviewers indicated that their concerns were addressed. Overall, the paper tackles an important and realistic problem with a coherent method and solid empirical support, while the remaining privacy concern is best viewed as a limitation to clarify rather than a flaw that overturns the contribution. Therefore, the AC recommends accepting the paper.